# The $^{226}$Ra-Ba relationship in the North Atlantic during GEOTRACES-GA01

Emilie Le Roy[1], Virginie Sanial[1,2,3], Matthew A. Charette[2], Pieter van Beek[1], François Lacan[1], Stéphanie H.M. Jacquet[4], Paul B. Henderson[2], Marc Souhaut[1], Maribel I. García-Ibáñez[5,6], Catherine Jeandel[1], Fiz F. Pérez[5], Géraldine Sarthou[7]

[1]LEGOS, Laboratoire d'Etudes en Géophysique et Océanographie Spatiales (Université de Toulouse, CNRS/CNES/IRD/UPS), Observatoire Midi Pyrénées, 14 Avenue Edouard Belin, 31400 Toulouse, France [emilie.le.roy@legos.obs-mip.fr]
[2]Department of Marine Chemistry and Geochemistry, Woods Hole Oceanographic Institution, Woods Hole, MA 02543, USA
[3]Now at Department of Marine Science, University of Southern Mississippi, Stennis Space Center, MS 39529, USA
[4]Aix Marseille Université, CNRS/INSU, Université de Toulon, IRD, Mediterranean Institute of Oceanography (MIO), UM110, 13288 Marseille, France
[5]Instituto de Investigaciones Marinas (IIM, CSIC), Eduardo Cabello 6, 36208 Vigo, Spain
[6]Now at Uni Research Climate, Bjerknes Centre for Climate Research, Bergen 5008, Norway
[7] Laboratoire des Sciences de l'Environnement Marin (LEMAR), UMR 6539, IUEM, Technopôle Brest Iroise, 29280 Plouzané, France

Correspondance to: Emilie Le Roy (*emilie.le.roy@legos.obs-mip.fr*)

**Abstract.** We report detailed sections of radium-226 ($^{226}$Ra, $T_{1/2}$ = 1602 y) activities and barium (Ba) concentrations determined in the North Atlantic (Portugal-Greenland-Canada) in the framework of the international GEOTRACES program (GA01 section—GEOVIDE project, May-July 2014). Dissolved $^{226}$Ra and Ba are strongly correlated along the section, a pattern that may reflect their similar chemical behavior. Because $^{226}$Ra and Ba have been widely used as tracers of water masses and ocean mixing, we investigated more thoroughly their behavior in this crucial region for thermohaline circulation taking advantage of the contrasting biogeochemical patterns existing along the GA01 section. We used an Optimum Multiparameter (OMP) analysis to distinguish the relative importance of physical transport (water mass mixing) and non-conservative processes (sedimentary, river, or hydrothermal inputs; uptake by particles, and dissolved-particulate dynamics) on the $^{226}$Ra and Ba distributions in the North Atlantic. Results show that the measured $^{226}$Ra and Ba concentrations can be explained by conservative mixing for 58 and 65 % of the samples respectively, notably at intermediate depth, away from the ocean interfaces. $^{226}$Ra and Ba can thus be considered as conservative tracers of water mass transport in the ocean interior at the space scales considered here, namely, of the order of few thousand km. However, regions where $^{226}$Ra and Ba displayed non-conservative behaviors and in some cases decoupled behaviors were also identified, mostly at the ocean boundaries (seafloor, continental margins, and surface waters). Elevated $^{226}$Ra and Ba concentrations found in deep waters of the West European Basin suggest that lower North East Atlantic Deep Water (NEADWl) accumulates $^{226}$Ra and Ba from

sediment diffusion and/or particle dissolution during transport. In the upper 1500 m of the West European Basin, deficiencies in $^{226}$Ra and Ba are likely explained by their incorporation in planktonic calcareous and siliceous shells, or in barite ($BaSO_4$) by substitution or adsorption mechanisms. Finally, because Ba and $^{226}$Ra display different source terms (mostly deep-sea sediments for $^{226}$Ra and rivers for Ba), strong decoupling between $^{226}$Ra and Ba were observed at the land-ocean boundaries. This is especially true in the shallow stations near the coasts of Greenland and Newfoundland where high $^{226}$Ra/Ba ratios at depth reflect the diffusion of $^{226}$Ra from sediment and low $^{226}$Ra/Ba ratios in the upper water column reflect the input of Ba associated with meteoric waters.

## 1. Introduction

The primary source of radium-226 ($^{226}$Ra, $T_{1/2} = 1602$ y) to the ocean was found to be the diffusion from deep-sea sediments following the decay of its parent isotope, $^{230}$Th (Koczy, 1958; Kröll, 1953). This mode of introduction led Koczy to use radium data to derive vertical eddy diffusivities and velocities in the deep sea (Koczy, 1958). Since then, $^{226}$Ra has been widely used to study the ocean circulation and mixing at a global scale (Chung and Craig, 1980; Ku et al., 1980). In the framework of the Geochemical Ocean Sections Study (GEOSECS) program, special attention was given to $^{226}$Ra as its solubility and half-life made it an ideal candidate as a chronometer of the global thermohaline circulation. In particular, its 1602 y half-life is more adapted than the longer half-life of carbon-14 ($^{14}$C, $T_{1/2} = 5730$ y) that had also been used for that purpose. Therefore, the global oceanic distribution of $^{226}$Ra in seawater was extensively documented throughout the major ocean basins and a unique database was generated during the GEOSECS program (Broecker et al., 1970, 1967; Chung et al., 1974; Ku et al., 1970; Ku and Lin, 1976).

Barium (Ba) is an alkaline earth element like $^{226}$Ra. Therefore they share a similar geochemical behavior in the ocean (Chan et al., 1976; Fanning et al., 1988; Mathieu and Wolgemuth, 1973). As such, Ba was proposed as a stable analog of $^{226}$Ra in order to use the $^{226}$Ra/Ba ratio as a clock in a similar manner as the $^{14}$C/$^{12}$C ratio. However, the recognition that $^{226}$Ra and Ba participate in upper ocean biological cycles (Ku and Luo, 1994) introduced additional complications for the use of the $^{226}$Ra/Ba ratio as a time tracer for deep water ventilation. Both $^{226}$Ra and Ba indeed increase with increasing depth, thus reflecting uptake due to biological processes in surface waters, particles scavenging and subsequent release at depth through the dissolution of the settling particles (Broecker et al., 1967; Ku et al., 1970; Ku and Luo, 1994). $^{226}$Ra and Ba are thus not only controlled by physical processes, but appear to be incorporated in settling particles such as calcareous and siliceous shells, or in barite ($BaSO_4$) that precipitates in the mesopelagic zone (Bishop, 1988; Chan et al., 1976; Dehairs et al., 1980; Lea and Boyle, 1989, 1989). Hence, despite different principal sources to the ocean, rivers in the case of

Ba and marine sediment diffusion for $^{226}$Ra, their distributions are affected by similar processes in the water column. Barium displays a linear correlation with $^{226}$Ra in the global ocean, resulting in a fairly constant $^{226}$Ra/Ba ratio of $2.2 \pm 0.2$ dpm µmol$^{-1}$ (dpm, disintegrations per minute) (Chan et al., 1976; Foster et al., 2004; Ku et al., 1980; Li et al., 1973; Östlund et al., 1987). Similarly, strong correlations were also found between Ba-Si (silicate) and $^{226}$Ra-Si although no obvious process links $^{226}$Ra or Ba with Si. These latter relationships appeared to be more surprising because Si is not a chemical analog of Ra and Ba. It was first proposed that diatom frustules exported from the upper water column could adsorb Ra and Ba, these elements being then released at depth following the dissolution of the siliceous tests (Bishop, 1988; Chung, 1980; Dileep Kumar and Li, 1996). More recent studies showed that the similar behaviors of Ba and Si (and alkalinity) reflect similar dissolved-particulate interactions (Jeandel et al., 1996; McManus et al., 1999; Rubin et al., 2003). Indeed, Ba is not mechanistically coupled with alkalinity or silicate. Rather, the observed relationships may result from the spatial coherence of different carriers (barite, opal and carbonate) overprinted by hydrodynamics. The formation of biogenic silica, CaCO$_3$ and barite in surface water and their subsequent dissolution in the deeper water column may thus generate parallel oceanic distributions. While barite has been shown to be the main carrier that controls the Ba water column distribution, the relationship between Ba-Ra remains unclear.

While the global GEOSECS program provided valuable information on the coupling between biogeochemical cycles of $^{226}$Ra and Ba in the ocean, several unknowns still remain. In this work, we take advantage of a new worldwide program, GEOTRACES, to provide new information on the distribution of $^{226}$Ra and Ba and their relationship in the ocean. GEOTRACES program aims to characterize the distribution of trace elements and their isotopes (TEIs) (sources, sinks, internal cycling) in the ocean through a global survey consisting of ocean sections and regional process studies.

In the present study, we report dissolved $^{226}$Ra activities and dissolved Ba concentrations in the North Atlantic Ocean and Labrador Sea (GEOVIDE project, GA01 section). The North Atlantic region hosts a variety of globally significant water masses with complex circulation patterns (García-Ibáñez et al., this issue; Lherminier et al., 2010). This area is crucial for the thermohaline circulation, and thus for global climate, through its important role in the ventilation of the deep layer of the global ocean (Seager et al., 2002). As part of this process, the Meridional Overturning Circulation (MOC) includes the northward transport of warm subtropical waters. These surface waters are then cooled and transformed into subpolar waters, and may reach the Labrador and Irminger Seas where deep-water formation and deep convection take place (Bennett et al., 1985; Pickart and Spall, 2007; Yashayaev et al., 2007). We propose to study the relationship between $^{226}$Ra and Ba and to test the conservative behavior of these tracers in this specific region. We further document the Ra-Ba-Si relationship along the GA01 section,

as it was done in previous sections conducted during the GEOSECS program and more recently along
the GEOTRACES GA03 section.
**2.  Materials and Methods**
**2.1. Study area; the GEOVIDE project**
The GEOTRACES GA01 section (GEOVIDE project; PIs : Géraldine Sarthou, LEMAR, France
and Pascale Lherminier, LOPS, France) was conducted in the North Atlantic Ocean between Lisbon,
Portugal, and St John's, Canada (15 May 2014-30 June 2014; Fig.1). The water samples described here
were collected on board the R/V *Pourquoi Pas?*. The section crossed different topographic features and
regions with contrasting biogeochemical patterns. It complemented the sections GA03 (U.S.-
GEOTRACES) and GA02 (Dutch GEOTRACES) also conducted in the Atlantic Ocean in the
framework of the GEOTRACES program. Seventy-eight stations were visited during the GEOVIDE
project.
**2.2. Sample collection**
At 15 of the 78 stations completed during the GA01 cruise, up to 22 discrete 10-L seawater samples
were collected through the water column from Niskin bottles. The seawater samples were passed by
gravity through 10 g of acrylic fibers impregnated with $MnO_2$ (so called "Mn fibers"), which
quantitatively adsorb radium isotopes (assumed to scavenge 100 % of Ra; van Beek et al., 2010; Moore
and Reid, 1973). High-resolution vertical profiles of $^{226}$Ra were thus built to provide a detailed $^{226}$Ra
section. The samples were unfiltered since particulate $^{226}$Ra activities are typically two orders of
magnitude lower than the dissolved $^{226}$Ra activities (van Beek et al., 2007, 2009). From the same Niskin
bottles, 15 mL was collected to determine the Ba concentration, so that Ba and $^{226}$Ra analyses were
conducted from the same initial sample, which allows us to investigate the $^{226}$Ra/Ba ratio in the
samples. The Ba samples were collected in pre-cleaned polypropylene bottles (rinsed three times with
the same seawater sample), acidified with 15 μL of HCl (10 M, Merck, Suprapur) and kept at room
temperature for later analysis.
**2.3. Analysis of dissolved $^{226}$Ra activities *via* $^{222}$Rn emanation**
Radium-226 was determined *via* its daughter, radon-222 ($^{222}$Rn; $T_{1/2}$ = 3.8 days) using a radon
extraction system followed by alpha scintillation counting (Key et al., 1979). The Mn Fiber samples
were placed into gas-tight PVC cartridges (Peterson et al., 2009) that were flushed with helium (He) for
5 min at 250 mL min$^{-1}$. The cartridges were sealed and held for approximately 2 weeks (minimum of 5
days) to allow for $^{222}$Rn ingrowth from $^{226}$Ra decay. The $^{222}$Rn was then flushed out from the cartridges
using He and cryo-trapped in copper tubing using liquid nitrogen. The copper trap was heated to allow
the $^{222}$Rn to be transferred to an evacuated "Lucas cells" *via* a stream of He. The "Lucas cells" are air-
tight chambers with inner walls coated with silver-activated zinc sulfide that emits photons when struck
by alpha decay particles (Key et al., 1979 ; Lucas, 1957 ; Peterson et al., 2009). The cells were held 3
hours to reach the secular equilibrium of all $^{222}$Rn decay chain daughters. After 3 hours, the samples
were counted overnight on a radon counting system (Model AC/DC-DRC-MK 10-2). The counting
uncertainties (1SD, Standard Deviation) were within the range of 2-5 % for 10 L volume samples. All
samples were appropriately ingrowth and decay corrected. The combined Lucas cell and detector
background was ~7 % of the typical total measured sample activity. The method was standardized using
NIST (U.S. National Institute of Standards & Technology) $^{226}$Ra (20 dpm) sorbed onto $MnO_2$ fiber and
analyzed in the same manner as the samples, with uncertainties (1SD) of 5 % (Charette et al., 2015;
Henderson et al., 2013). Vertical profiles of $^{226}$Ra from the GEOTRACES GA01 (this study) and GA03
(Charette et al., 2015) sections that were located in close proximity off Portugal (Fig.1) were compared,
and showed a good agreement with increasing activities with increasing depth (Fig.S1).
**2.4. Analysis of dissolved Ba concentrations**
Barium concentrations were measured using an isotope dilution (ID) method (Freydier et al., 1995;
Klinkhammer and Chan, 1990) by high resolution-inductively coupled plasma-mass spectrometry (HR-
ICP-MS). This method was adapted to a Thermo Finnigan Element XR instrument (MIO, Marseille).
The Ba measurements presented here are the sum of dissolved Ba and a very small fraction of
particulate Ba (generally <1 % of total Ba, along GEOVIDE up to 1.3 % at the bottom of station 32 due
to presence of a nepheloid layer; Lemaitre et al., this issue) released from the samples as a result of the
acidification step. Hence, while the measurements reported herein are total Ba, they are within
analytical uncertainty representative of the dissolved Ba pool. The samples (0.5 mL) were spiked with
300 µL of a $^{135}$Ba-enriched solution (93 % $^{135}$Ba; 95 nmol kg$^{-1}$) and diluted with 15 mL of acidified
(2 % $HNO_3$, 14 M, Optima grade) Milli-Q grade water (Millipore). The amounts of sample, spike and
dilution water were assessed by weighing. The reproducibility of this method is about 1.5 % (1 RSD,
Relative Standard Deviation), as tested on repeated preparations of the reference solution SLRS-5
(NRC-CNRC river water reference material for trace metals). Average Ba values obtained for SLRS-5
were 13.48 ± 0.20 µg L$^{-1}$ (1 σ) with RSD of 1.5 %, which is in good agreement with the certified values
(SLRS-5 13.4 ± 0.6 µg L$^{-1}$). The limit of detection calculated as three times the standard deviation of the
procedural blank was 0.09 nmol L$^{-1}$.

**2.5. Multiparameter mixing model**

An Optimum MultiParameter (OMP) analysis was used to distinguish the relative importance of physical transport (i.e., water mass mixing) from non-conservative processes (input from the sediments, rivers or hydrothermal vents, dissolution of particles; uptake by particles and dissolved-particulate dynamics) on the $^{226}$Ra and Ba distributions in the North Atlantic. We used the OMP analysis computed for the GA01 section by Garcia Ibanez et al. (this issue) with 12 source water types (SWTs). Based on historical data reported from the North Atlantic, we defined $^{226}$Ra and Ba endmember concentrations associated with each SWT (Table S1). The characteristics of SWTs (potential temperature, salinity, and geographical location) reported by Garcia Ibanez et al. (this issue) were used to determine the SWTs endmembers for $^{226}$Ra and Ba,). In some cases, data from the GA01 section were used for the SWT endmember (Table S1). These $^{226}$Ra and Ba SWT endmembers were then used to calculate the $^{226}$Ra and Ba concentrations that strictly result from mixing of the different water masses. In this way, we estimated the conservative component of $^{226}$Ra and Ba, which can in turn be compared to the *in situ* concentrations to generate the non-conservative component of $^{226}$Ra and Ba along the GA01 section.

The uncertainties for the SWT endmembers were considered to be $\pm$ 0.6 dpm 100 L$^{-1}$ for $^{226}$Ra and $\pm$ 1.5 nmol L$^{-1}$ for Ba, based on typical measurement uncertainties and spatial variability. A Monte Carlo method (250 random perturbations) was used to propagate the SWT endmember uncertainties to the $^{226}$Ra and Ba results of the OMP analysis. This allowed us to estimate conservative component uncertainties of $\pm$ 0.9 dpm 100 L$^{-1}$ for $^{226}$Ra and $\pm$ 4.4 nmol L$^{-1}$ for Ba. When the measured $^{226}$Ra and Ba values were different from the conservative component values (taking into account the uncertainties on these values), $^{226}$Ra and Ba were considered as non-conservative. These non-conservative values can either be positive (representative of a net addition of $^{226}$Ra and Ba) or negative (representative of a net removal of $^{226}$Ra and Ba).

Note that the OMP analysis was not solved where non-conservative behavior of temperature and salinity is expected (that is, for waters above 100 m and for waters with salinities lower than 34.7). In these latter cases, changes in water mass properties may indeed be due to air-sea interaction or inputs of fresh waters (e.g., near Greenland shelf; Daniault et al., 2011).

**3. Results**

**3.1. Hydrodynamic context**

The OMP analysis was used to identify the different water masses (Table 1) crossing the GA01 section. The potential temperature-salinity diagram for all the GA01 stations along with the different

SWT endmembers used in the OMP analysis are represented in Fig.2. The salinity section is shown in Fig.3. The different water masses present along the GA01 section are described below.

### 3.1.1.  Upper waters

Three main water masses were found in upper waters (<1000 m) in the investigated area (Fig.3). First, the Central Waters occupied the upper eastern part of the GA01 section from the Iberian Peninsula to the Rockall Through (stations 1 to 26). Their distribution was associated with the circulation of the North Atlantic Current (NAC). The NAC transports warm and saline waters northward, connecting the subtropical and the subpolar latitudes, and is part of the upper layer of the Atlantic Meridional Overturning Circulation (AMOC) in the North Atlantic subpolar gyre. The NAC flows eastward from the Grand Banks of Newfoundland, splitting into four branches west of the Mid-Atlantic Ridge (MAR), while incorporating local water masses (Fig.1). East of the MAR, the two northern branches of the NAC flow northward into the Icelandic Basin, the Rockall Plateau and the Rockall Trough, while the two southern branches flow southward into the West European Basin. The Central Waters can be identified by the highest potential temperature of the entire GA01 section and are represented by two endmembers called East North Atlantic Central Waters ($ENACW_{16}$ and $ENACW_{12}$). The $ENACW_{16}$ is warmer (16 °C) than the $ENACW_{12}$ that can be identified with a potential temperature of 12.3 °C (Fig.2).

Part of the Central Waters carried by the NAC recirculates toward the Iceland Basin and the Irminger Sea, leading to the formation of subpolar mode waters by mixing and cooling in the subpolar gyre (Lacan and Jeandel, 2004; McCartney, 1992). Iceland Subpolar Mode Water (IcSPMW) is formed in the Icelandic Basin and is located, along GA01, over the Reykjanes Ridges (stations 32 and 38) (Fig.3). The IcSPMW is described by two endmembers, the $SPMW_7$ and the $SPMW_8$, which are distinguished by their potential temperature of 7.0 °C and 8.0 °C, respectively (Fig.2). Once formed, the IcSPMW follows the Irminger Current.

Finally, the Irminger Subpolar Mode Water (IrSPMW) is the result of the transformation of the Central Waters and the IcSPMW, and is formed northwest of the Irminger Sea (Krauss, 1995). The IrSPMW is located near Greenland (stations 53, 57 and 60) (Fig.3) (García-Ibáñez et al., 2015; Lacan and Jeandel, 2004; Read, 2000).

### 3.1.2.  Intermediate waters

The Subarctic Intermediate Water (SAIW) originates in the Labrador Current (Read, 2000). The SAIW is associated with the advection of waters from the Labrador Sea within the NAC; it subducts below the Central Waters at approximately 600 m. Low salinities (34.8 and 34.7) and potential

temperatures of 4.5 °C and 6 °C are representative of the two SAIWs, $SAIW_4$ and $SAIW_6$, respectively (Fig.2).

Around the Rockall Plateau, the SAIW overlies the Mediterranean Water (MW). The MW enters the North Atlantic through the Gibraltar Strait and flows northward while extending westward. The MW can be identified in the West European Basin at approximately 1200 m (stations 1 and 13 in Fig.3) with a salinity of 36.5 (Fig.2; García-Ibáñez et al., 2015).

The Labrador Sea water (LSW) is found in multiple locations and at different water depths along the GA01 section (Fig.3). The LSW is formed by progressive cooling and freshening in winter due to deep convection. The LSW can be characterized by its minimum salinity (34.87) and its minimum potential temperature (3 °C) (Fig.2). The LSW contributes to the stratification of the interior of the North Atlantic and its boundary currents and spreads at intermediate depths in three different basins intersected by the GA01 section (Fig.1). The three independent pathways are: (i) northward into the Irminger Sea (station 44), (ii) eastward across the MAR, through the Charlie-Gibbs fracture zone, then northward into the Iceland Basin (station 32) and eastward into the West European Basin (stations 21 and 26), and (iii) equatorward as a major component of the North Atlantic Deep Water in the Deep Western Boundary Current (DWBC), which constitutes the lower limb of the AMOC. Along these paths, the LSW mixes with both the overlying and underlying water masses and becomes warmer and saltier (Lazier, 1973).

The Polar Intermediate Water (PIW) is characterized by very low salinity (34.9) and potential temperature (less than 2 °C) (Fig.2) and is defined as a separate upper core on the Greenland slope. The PIW is episodically injected into the Irminger Sea and originates from either the Arctic Ocean or the Greenland shelf (Falina et al., 2012; Jenkins et al., 2015; Rudels et al., 2002).

### 3.1.3. Overflow waters and deep waters

The Iceland-Scotland Overflow Water (ISOW) originates at the Iceland-Scotland sill, and entrains the overlying warm saline atlantic waters (SPMW and LSW). ISOW identification features are a potential temperature of 2.6 °C and a salinity of 34.98 (Fig.2; van Aken and Becker, 1996). ISOW was found at stations located on the Eastern flank of the Reykjanes Ridge (stations 32 and 38) and near Greenland (stations 60 and 64) at great depth (2000-3500 m) (Fig.3).

Overflow waters coming from the Denmark Strait mix with both the SPMW and the LSW during descent into the Irminger Sea to form the Denmark Strait Overflow Water (DSOW) (Fig.1) (Read, 2000; Yashayaev and Dickson, 2008). DSOW is located at the northern end of the Irminger Sea (station 44) and occupies the deepest part of the Greenland continental slope (stations 69 and 77)

(Fig.3). At bottom depth, DSOW is easily identified by a minimum potential temperature of 1.3 °C
(Fig.2).
In the Southern Ocean, the Antarctic Bottom water (AABW) is formed by deep winter convection
of surface waters. AABW flows to the north along the eastern side of the Atlantic and contributes to the
formation of the lower North East Atlantic Deep Water (NEADWl) once this water penetrates the
Iberian Abyssal Plain by crossing the Mid-Atlantic Ridge (Fig.1). The NEADWl is laying at the bottom
of the West European Basin (stations 1 to 26 in Fig.3) with a mean salinity of 34.895 and a potential
temperature of 1.98 °C (Fig.2). Then, the NEADWl recirculates into the Rockall Trough and meets
ISOW in the Iceland Basin (van Aken, 2000; McCartney, 1992; Schmitz and McCartney, 1993).
**3.2. Distribution of $^{226}$Ra and Ba along the GA01 section**
The $^{226}$Ra distribution for the GA01 section is presented in Fig.4 with Ba concentrations and
$^{226}$Ra/Ba ratios. The $^{226}$Ra activities and Ba concentrations in the water column range from 7 to
25 dpm 100 L$^{-1}$ and from 33.6 to 81.5 nmol L$^{-1}$, respectively. These data are in good agreement with
Atlantic data from the GEOSECS program, which range from 6.8 to 23.4 dpm 100 L$^{-1}$ for $^{226}$Ra and
from 35 to 105 nmol L$^{-1}$ for Ba (Broecker et al., 1976).
For both $^{226}$Ra and Ba, the vertical gradient is stronger in the eastern part of the section (West
European Basin) than on the western part of the section (from Reykjanes Ridge to Newfoundland). Both
are particularly high near the seafloor in the West European Basin ($^{226}$Ra: 14-25 dpm 100 L$^{-1}$; Ba:
63-82 nmol L$^{-1}$) and are in agreement with data previously reported for this region (Broecker et al.,
1976; Charette et al., 2015). At intermediate depths, Ba concentrations range from 40 to 50 nmol L$^{-1}$ in
the West European Basin (stations 1 and 21) and $^{226}$Ra activities range from 9.5 to 10.9 dpm 100 L$^{-1}$,
also in good agreement with literature data (Charette et al., 2015; Schmidt and Reyss, 1996). Low $^{226}$Ra
and Ba are found in upper waters of the West European Basin and the Iceland Basin
(8.1-8.9 dpm 100 L$^{-1}$ and 35-43 nmol L$^{-1}$, respectively). Intermediate $^{226}$Ra activities and Ba
concentrations (9 dpm 100 L$^{-1}$ and 42 nmol L$^{-1}$, respectively) are observed in bottom waters in Irminger
and Labrador Seas. Between the Reykjanes Ridge and Newfoundland, $^{226}$Ra activities range between 7-
10 dpm 100 L$^{-1}$ in surface and intermediate waters. Similar to $^{226}$Ra, Ba concentrations are relatively
low in this area, ranging from 39-50 nmol L$^{-1}$. The distributions in $^{226}$Ra and Ba are to a first order
explained by the different water masses present in the region, as discussed below.

# 4. Discussion

## 4.1. $^{226}$Ra-Ba and $^{226}$Ra-Ba-Si relationships

A linear correlation between $^{226}$Ra and Ba is observed for all data collected along the GA01 section (Fig.5). The slope of the $^{226}$Ra-Ba linear regression obtained by this study in the North Atlantic is $2.5 \pm 0.1$ (2SD) dpm µmol$^{-1}$ which agrees with the slope of the $^{226}$Ra-Ba linear regression of 2.3 dpm µmol$^{-1}$ reported during the GEOSECS program for all the oceanic basins (Chan et al., 1976). The intercept on the horizontal Ba axis is 4.4 nmol L$^{-1}$ for the GA01 section, which is in agreement with GEOSECS data (Chan et al., 1976; Li et al., 1973). This positive intercept may be the result of a larger riverine Ba input relative to $^{226}$Ra (Ku and Luo, 1994). The slope of the $^{226}$Ra-Ba linear regression reported from the GEOSECS program is similar from one oceanic basin to the other. The $^{226}$Ra/Ba ratio (slightly different from the slope) is also fairly constant throughout the global ocean ($2.2 \pm 0.2$ dpm µmol$^{-1}$; Östlund et al., 1987). This pattern indicates that $^{226}$Ra and Ba may behave similarly in the ocean. Since $^{226}$Ra and Ba are incorporated in settling particles such as calcareous and siliceous shells or barite (BaSO$_4$) by substitution or adsorption mechanisms (Bishop, 1988; Dehairs et al., 1980; Lea and Boyle, 1989, 1990) and are then released at depth following the dissolution of these particles, the constant $^{226}$Ra/Ba ratio suggests that fractionation between $^{226}$Ra and Ba during these processes is small.

Investigations conducted during the GEOSECS program further concluded that $^{226}$Ra and Ba were tightly correlated to orthosilicic acid (Si(OH)$_4$) (Chan et al., 1976; Chung, 1980; Ku et al., 1970; Ku and Lin, 1976) despite the fact that $^{226}$Ra, Ba, and Si(OH)$_4$ exhibit different chemical behavior in the water column and different source terms. A Ra-Ba-Si relationship is also observed along the GA01 section (Fig.5). Si(OH)$_4$ concentrations generally increase with increasing depth, with a steeper gradient in the West European Basin (Introduction Paper, 2017; This issue), as it was also the case for $^{226}$Ra and Ba (Fig.S2). The link between $^{226}$Ra, Ba and Si has been shown to reflect parallel dissolved-particulate interactions between barite and biogenic silica (Bishop, 1988; Chung, 1980; Jacquet et al., 2005, 2007; Jeandel et al., 1996); the main carrier of $^{226}$Ra in the ocean, however, remains an open question. The oceanic Ba-Si and $^{226}$Ra-Si relationships may thus be the result of the interaction between ocean biogeochemistry and the water mass transport.

In contrast to the $^{226}$Ra-Ba relationship, the slope of the $^{226}$Ra-Si(OH)$_4$ relationship observed during GEOSECS exhibited significant spatial variability from one oceanic basin to the other (Li et al., 1973). First, $^{226}$Ra and Si are not chemical analogues, as it is the case for $^{226}$Ra and Ba. Second, the variability observed in the $^{226}$Ra-Si(OH)$_4$ relationship may be related to the large variability of the Si(OH)$_4$ distribution which is mostly governed by the preformed nutrient concentrations of waters feeding into

the main thermocline from surface waters of the Southern Ocean (Sarmiento et al., 2007). In the case of GA01, the $^{226}$Ra-Si(OH)$_4$ linear regression slope is 2.4 ± 0.9 (2SD) 10$^3$ dpm mol$^{-1}$. As a comparison, the $^{226}$Ra-Si(OH)$_4$ slope reported for the GEOTRACES-GA03 section conducted south of the GA01 section in the Atlantic Ocean was 2.1 10$^3$ dpm mol$^{-1}$ (Charette et al., 2015). The slope of the $^{226}$Ra-Si(OH)$_4$ linear regression is 34.3 10$^3$ dpm mol$^{-1}$ in the Pacific Ocean and 14.5 10$^3$ dpm mol$^{-1}$ in the Antarctic Ocean. The $^{226}$Ra-Si(OH)$_4$ relationship has an intercept with the vertical axis of 7.3 ± 0.1 dpm 100 L$^{-1}$, which represents the residual $^{226}$Ra resulting from the total usage of Si in surface waters (Ku et al., 1970). According to (Shannon and Cherry, 1971), the removal of $^{226}$Ra in the upper waters is limited by Si. For both the $^{226}$Ra-Ba and $^{226}$Ra-Si(OH)$_4$ relationships, several values are clearly outside of the linear regression trend (Fig.5), a pattern that indicates deviation from the relationship usually observed between $^{226}$Ra and Ba (or Si(OH)$_4$). Such deviations may result from non-conservative processes.

## 4.2. $^{226}$Ra and Ba distributions and their relationship with hydrography

A striking feature of the GA01 section is that the $^{226}$Ra activities and Ba concentrations are particularly high in the West European Basin below 2000 m (Fig.4), in the NEADWl. This pattern can also be observed in the GA03 section conducted south of the GA01 section (Charette et al., 2015), the two sections being separated by only ca. 500 km in that basin. The NEADWl is mainly formed from waters with a southern origin (Read, 2000). South of the Antarctic Convergence, the surface waters contain high $^{226}$Ra activities from the upwelling of deep waters enriched in $^{226}$Ra associated with the circumpolar current (Ku and Lin, 1976). The convection of these surface waters leads to the formation of the $^{226}$Ra-rich AABW that circulates northward into the Atlantic Ocean. However, the high $^{226}$Ra activities of the NEADWl cannot be solely explained by the high $^{226}$Ra activities of these waters of southern origin. This will be discussed in section 4.3.1.

In contrast, the lowest $^{226}$Ra activities and Ba concentrations reported on the GA01 section are associated with the Central Waters (upper waters of the West European Basin; Fig.4). Central Waters are derived from the NAC and mix with the SAIW and the SPMW. Along their path, Central Waters remain in the upper water column, and therefore are not affected by the deep sedimentary source of $^{226}$Ra. West of the Iceland Basin between 200 and 400 m (stations 32 and 38), an increase in the $^{226}$Ra activities and Ba concentrations is associated with the IcSPMW.

A slight increase in $^{226}$Ra is observed between 1000-1600 m in the West European Basin (Fig.4; Stations 1 and 13) where a salinity maximum is identified. This pattern is associated with the MW. This is corroborated by the slightly higher Ba concentrations and lower $^{226}$Ra/Ba ratios, as observed in the Western Mediterranean Sea (van Beek et al., 2009), these waters spreading westward into the Atlantic

Ocean. At these stations, between 30 and 79 % of the water found at 1000-1600 m is of Mediterranean origin (MW), according to the OMP analysis (Garcia Ibanez et al., 2018; this issue).

Relatively uniform and low $^{226}$Ra activities and Ba concentrations are found between the surface and 2500 m in the Labrador Sea, up to 2000 m in the Iceland Basin and deeper in the Irminger Basin (Fig.4). These distributions can be related to the LSW which is formed by winter convection in the Labrador Sea (Fröb et al., 2016; Pickart et al., 2003; Yashayaev and Loder, 2016). When formed, the LSW transports the characteristics of surface waters to the deep ocean (i.e., low $^{226}$Ra activities and low Ba concentrations). The LSW then spreads into the Irminger and the Iceland Basin while conserving its low $^{226}$Ra and Ba signatures. Relatively low $^{226}$Ra activities and Ba concentrations are found at bottom depths in the Irminger and Labrador Seas and may be associated with DSOW, which is also a recently ventilated water mass (Lazier, 1973).

Finally, according to the OMP analysis, ISOW is present at several stations along the GA01 section (Garcia-Ibanez et al., this issue). First, on the eastern flank of the Reykjanes Ridge (station 32), 68 % of the water between 2700 and 3000 m is considered as ISOW. Then, in the Labrador Sea (stations 69 and 77), an average of 58 % of the water between 2100 and 3000 m is identified as ISOW. The slight increase in $^{226}$Ra activities and Ba concentrations observed at these locations may thus be related to the ISOW.

## 4.3. Conservative versus non-conservative behavior of $^{226}$Ra and Ba

The use of an Optimum Multiparameter (OMP) analysis allowed us to distinguish the relative importance of physical transport (i.e., mixing of water masses) from non-conservative processes on the $^{226}$Ra, Ba and $^{226}$Ra/Ba ratios distributions in the North Atlantic (Fig.6). The comparison between the vertical profiles of $^{226}$Ra and Ba determined along the GA01 section, and those derived from OMP analysis (Fig.7) clearly indicates deviations from the conservative behavior and reflects either an input of $^{226}$Ra or Ba (positive anomalies highlighted in red; same color code as in Fig.6) or a removal of $^{226}$Ra or Ba (negative anomalies highlighted in blue; same color code as in Fig.6). This comparison reveals that for 58 % of the samples $^{226}$Ra can be considered as conservative (activities due to mixing and transport) along the GA01 section (i.e., 58 % of the $^{226}$Ra anomalies are within the [-0.9 and 0.9 dpm 100 L$^{-1}$] interval), whereas for 65 % of samples Ba can be considered as conservative (i.e., 65 % of the Ba anomalies are within the [-4.4 and 4.4 nmol L$^{-1}$] interval). A major finding of this study is that $^{226}$Ra and Ba are predominantly conservative at intermediate depths: mostly between 500 m to 2000 m, but slightly deeper in the center of deep basins such as at stations 21, 44 and 69 (Fig.6). These locations correspond to the depths where the waters are far from the main sources and sinks of $^{226}$Ra and Ba. The non-conservative $^{226}$Ra (42 % of the $^{226}$Ra) is mostly distributed close to the interfaces such

as surface/subsurface waters and bottom waters (both in the deep West European Basin and the
Labrador Sea), near the main sources (seafloor or shallow sediments deposited onto the margins). The
non-conservative Ba is mostly distributed in the upper 1500 m and in the deep West European Basin
(Fig.6).
The $^{226}$Ra/Ba ratios are also reported for all samples in Fig.7. The mean ratio determined along the
GA01 section is identical to the ratio determined during the GEOSECS program (2.2 ± 0.2 dpm µmol$^{-1}$;
Östlund et al., 1987). 77 % of the $^{226}$Ra/Ba ratios determined along the GA01 section are within the
confidence interval [2.0-2.4 dpm µmol$^{-1}$], indicating little deviation from the mean ratio, a pattern that is
likely related to the similar chemical behavior between $^{226}$Ra and Ba.

## 4.3.1.  $^{226}$Ra inputs and their relationship with Ba

Deep waters of the West European Basin display positive $^{226}$Ra and Ba anomalies (stations 1 to
26; Fig.7). The $^{226}$Ra anomalies are initiated at shallower depths (ca. 300-2000 m) than the Ba
anomalies (ca. 1000—2000m) (Fig.7). As mentioned above, the NEADWl—which is the main water
mass of the deep West European Basin—is mainly formed from waters with a southern origin (mainly
AABW) that are characterized by high $^{226}$Ra and Ba concentrations. However, these southern waters
experience a very specific history along their northward transport to the GA01 section that suggests that
the high $^{226}$Ra activities (and Ba) of the NEADWl cannot be solely explained by the high $^{226}$Ra
activities (and Ba) of these waters of southern origin. In order to explain the positive $^{226}$Ra and Ba
anomalies in the deep waters of the West European Basin, we thus need to investigate the fate of $^{226}$Ra
and Ba in the waters of southern origin that travel northward and reach section GA01. Figure 8 was
computed combining GEOSECS and TTO data ($^{226}$Ra, Si(OH)$_4$, salinity and temperature) associated
with the AABW (Spencer, 1972) that travels northward between 60 °S and 40 °N in the West Atlantic
Basin. The same data ($^{226}$Ra, Si(OH)$_4$, salinity and temperature) associated with the NEADWl in the
East Atlantic Basin and along GA01 are also reported.
Between 60 °S and the equator, the high $^{226}$Ra activities of the AABW are associated with
relatively low salinity, low temperature, and high Si(OH)$_4$ (Fig.8). Then, while crossing the Mid-
Atlantic Ridge at the equator and at 11 °N, the AABW undergoes several important transformations:
$^{226}$Ra activities and Si(OH)$_4$ concentrations decrease while salinity and temperature tend to increase
(Fig.8). The $^{226}$Ra and Ba endmembers for the NEADWl were chosen at this specific location to
coincide both geographically and with the characteristics (salinity, temperature and Si(OH)$_4$) of the
NEADWl endmembers used for the OMP analysis (Fig.8; Fig.S3). This endmember has been chosen far
from the GA01 section in the OMP analysis (García-Ibáñez et al., this issue), because between 11 °N
and the GA01 section (Fig.8), salinity, temperature, and Si(OH)$_4$ concentrations display relatively

constant trends indicating no major modifications. In contrast, the [226]Ra activities display a significant spatial variability north of 11 °N, and clearly increase towards the north (Fig.8). This [226]Ra increase is thus decoupled from salinity, temperature, and $Si(OH)_4$), and likely explains the positive anomalies deduced from the OMP analysis in the deep West European Basin (Fig.7). The specific history of these waters of southern origin (waters initially with a high [226]Ra activity; decrease in the [226]Ra activity at the equator and at 11 °N; new increase of [226]Ra activity north of 11 °N) suggest that the [226]Ra anomalies observed in the West European Basin are explained by inputs of [226]Ra along the northward transport of these waters.

The positive anomalies result from the input of [226]Ra (and Ba) following either i) dissolution/remineralization of settling particles that incorporated [226]Ra and Ba in the upper water column (this includes the dissolution of barite, since the waters of the Atlantic Ocean are undersaturated with respect to barite; Monnin et al., 1999) and/or ii) diffusion of [226]Ra and Ba from deep-sea sediments (Cochran and Krishnaswami, 1980) (see 4.4). Of special note are stations in the West European Basin, which are especially deep (down to 5500 m). Deep sediments generally display elevated [230]Th activities due to scavenging of [230]Th from the entire water column (Bacon and Anderson, 1982; Nozaki, 1984). The highest dissolved [230]Th activities reported along the GA01 section were thus observed in the deep waters of the West European Basin (Deng et al., 2017, this issue). Consequently, because [226]Ra is produced by the decay of [230]Th in the sediment, the [226]Ra diffusive flux in this area is expected to be especially high.

The input of [226]Ra in the West European Margin is accompanied by a Ba input since i) positive Ba anomalies are also observed in the deep waters and ii) the [226]Ra/Ba ratios do not significantly deviate from the mean GEOSECS [226]Ra/Ba ratio (Fig.7a). One exception is found at station 21 in the West European Basin, which displays high [226]Ra/Ba at approximately 4000 m (up to 3.17 dpm µmol[-1]). At several stations (21, 26, 32, 38, 44, 60, 64 and 77), lower beam transmission values near the seafloor indicate presence of suspended sediments likely associated with a nepheloid layer. Nepheloid layers are turbid layers formed episodically by strong and intense abyssal currents that are transported along isopycnals away from the site of resuspension of bottom sediments (McCave, 1986). The presence of a benthic nepheloid layer is also indicated by high particulate iron concentrations at these stations (Gourain et al., 2017; this issue). Such process may thus contribute to release [226]Ra (and potentially Ba) to the deep water column, following desorption or dissolution of the particles. Similar [226]Ra maxima have been observed in the deep waters of the Northeast Pacific suggesting that the [226]Ra flux is not uniform over the ocean bottom even on a regional scale (Chung, 1976). Suspended particle dissolution may also play a role here, notably for Ba. This will be discussed in more detailed in section 4.4.

Positive $^{226}$Ra anomalies are also found in deep waters at several other stations located in relatively deep basins (> 1200 m) along the GA01 section (e.g. stations 32, 38, 44, 60, 64, 69 and 77). Most of these anomalies are associated with $^{226}$Ra/Ba ratios higher than 2.4 dpm µmol$^{-1}$. The $^{226}$Ra positive anomalies observed at the stations mentioned above are thus best explained by the diffusion of $^{226}$Ra from the sediment. However, these latter stations do not exhibit a positive Ba anomaly and Ba tends to be conservative. Consequently, the $^{226}$Ra/Ba ratios in the deep waters of these stations are often significantly higher than the mean GEOSECS value (stations 21, 32, 38, 60, 64; Fig.7). This pattern is different to that observed in the West European Basin, a discrepancy that may be explained by the different sediment composition in the two regions, by the different residence time of deep waters in contact with deep-sea sediments (Chung, 1976) and/or different role played by suspended particles dissolution.

A strong $^{226}$Ra positive anomaly is observed in the deepest sample collected at station 38 above the Reykjanes Ridge. It cannot be completely excluded that this positive anomaly is attributed to hydrothermal vent since hydrothermal systems are known in the area (Fig.1). High particulate iron and aluminum concentrations were also observed at these stations (Gourain et al., 2017 ; Menzel et al., 2017 ; this issue). Enrichment in $^{226}$Ra have indeed been observed in hydrothermal systems plume at mid-ocean Ridges (Kadko, 1996; Kadko and Moore, 1988; Kipp et al., 2017; Rudnicki and Elderfield, 1992). Moreover, the $^{226}$Ra enrichments are accompanied by slight Ba enrichments, which may support the hydrothermal origin hypothesis, since hydrothermal venting at mid-ocean Ridge constitutes the second major external source of Ba to the ocean (Edmond et al., 1979).

Finally, high $^{226}$Ra/Ba ratios variations are also observed in shallow coastal waters (Fig.7c). At stations 53 and 61, high $^{226}$Ra/Ba ratios are found close to the bottom, in agreement with the input of $^{226}$Ra from the sediment, whereas low $^{226}$Ra/Ba ratios are found in subsurface at stations 57, 61 and 78, in association with low salinities (Fig.S2). The low $^{226}$Ra/Ba ratios are thus explained by the input of meteoritic waters in coastal areas, since such waters are known to be the predominant source of Ba to the ocean (Martin and Meybeck, 1979; Wolgemuth and Broecker, 1970). At these shallow stations, the different source terms between $^{226}$Ra and Ba therefore leads to important variations of the $^{226}$Ra/Ba ratios (Fig.7c.; stations 53, 57, 61 and 78). These observations clearly indicate that $^{226}$Ra may sometimes be decoupled from Ba.

## 4.3.2. $^{226}$Ra removal and its relationship with Ba

Relatively few $^{226}$Ra negative anomalies were found along the GA01 section. At the deep open-ocean stations, the negative anomalies are mostly observed in the upper 1000 m (Fig.7; stations 13, 21, 26, 32, 38, 44 and 77), but can also be found as deep as 2000 m (i.e., stations 64 and 69). In most cases,

the negative $^{226}$Ra anomalies are associated with significant negative Ba anomalies (stations 13,21, 26, 38, 44, 64 and 69). Such features are likely explained by biological mediated processes including incorporation of $^{226}$Ra and Ba in planktonic as calcareous and siliceous shells (Bishop, 1988), in acantharian skeletons made of celestite ($SrSO_4$) or in barite ($BaSO_4$) crystals (van Beek et al., 2007; Chow and Goldberg, 1960; Shannon and Cherry, 1971; Szabo, 1967; Wolgemuth and Broecker, 1970).

Particularly low dissolved $^{226}$Ra/Ba ratios (<2 dpm µmol$^{-1}$) are found in the upper 50 m at stations 21, 32, 64, 69 and 77, a pattern that was also observed in the upper 150 m of the Sargasso Sea, where van Beek et al., (2007) reported similar low dissolved $^{226}$Ra/Ba ratios that were accompanied by high $^{226}$Ra/Ba ratios in suspended particles. This latter pattern was attributed to the incorporation of $^{226}$Ra and Ba in acantharian skeletons. The low dissolved $^{226}$Ra/Ba ratios (e.g. 1.7 dpm µmol$^{-1}$, station 77) observed in the upper 200 m along the GA01 section may thus be attributed to acantharians, which have skeletons that incorporate $^{226}$Ra preferentially to Ba (van Beek et al., 2007, 2009; Bernstein et al., 1998). The presence of Acantharians was not studied during GEOVIDE. However, previous studies reported presence of acantharians in this area, like for example in the Iceland Basin and in the East Greenland Sea (Antia et al., 1993; Barnard et al., 2004; Martin et al., 2010).

Several phytoplankton blooms were observed along the GA01 section, as indicated by the chlorophyll a concentrations (Chl-a) (Fig.S4). The largest bloom was observed in the Labrador Sea in May 2014. Diatoms were the dominant species in the Irminger and Labrador Seas and on the Greenland and Newfoundland margins during GA01 (up to 55 % of the total Chl-a concentration; Tonnard et al., in prep). In the West European Basin, Chl-a was lower in May and June 2014 and coccolithophorids were the dominant species in that area (Tonnard et al., in prep). In these two regions, diatom frustules and coccolithophorids may thus contribute to the removal of $^{226}$Ra and Ba (Bishop, 1988; Dehairs et al., 1980), from the water column in these areas that were characterized by noticeable negative anomalies.

Additionally, because the Labrador Sea was sampled in June, during the decline of the bloom, barite that is presumably formed following the decay of settling organic matter may also contribute to the removal of Ba (and $^{226}$Ra). High particulate excess Ba (Ba$_{xs}$) concentrations were indeed observed at stations displaying significant Ba negative anomalies: Ba$_{xs}$ concentrations reached a maximum at 400 m at station 13, 21 and 26) and between 400 and 800 m near Greenland, at stations 44, 64 and 69 (Lemaitre et al., 2017, *Ba paper,* this issue). These Ba$_{xs}$ profiles can be related to the relatively high particulate organic carbon (POC) export flux determined at these stations (e.g. at station 69, Lemaitre et al., 2017, *Export paper;* this issue). This POC flux would promote barite formation in subsurface (Dehairs et al., 1980; Legeleux and Reyss, 1996) but also deeper in the water column (van Beek et al., 2007). Similarly, Jullion et al., (2017)—by using a parametric OMP analyses as applied in the Mediterranean Sea—also reported quantification of the non-conservative component of the Ba signal

and suggested that the Ba negative anomalies potentially reflected Ba subtraction during barite formation occurring during POC remineralization. The winter deep convection in the Labrador Sea may also potentially explain this relatively deep Ba anomalies by transporting negative Ba and $^{226}$Ra anomalies waters toward the deep-sea (Jullion et al., 2017). With the exception of the acantharian skeletons that may impact the dissolved $^{226}$Ra/Ba ratios in the upper 200 m, the removal of $^{226}$Ra and Ba that takes place deeper in the water column or that involves other processes (e.g. barite precipitation) does not seem to affect significantly the dissolved $^{226}$Ra/Ba ratios.

In the shallow coastal stations, lower $^{226}$Ra/Ba ratios are observed (Fig.7). These low ratios may also result from the removal of $^{226}$Ra and Ba by planktonic shells and/or barite or scavenging by lithogenic particles. However, because these stations are coastal stations, various processes are at play in a relatively shallow water column (i.e. diffusion of $^{226}$Ra from the sediments; input of Ba from meteoritic water; removal of Ba and $^{226}$Ra by shells and barite) thus complicating the interpretation of the vertical profiles. We note that the low $^{226}$Ra/Ba ratios observed in surface of shallow stations near the coast of Greenland (stations 57 and 61) and Newfoundland (station 78) are associated with low salinities (Fig.7c). This decoupling between $^{226}$Ra and Ba may be explained by input of freshwater into the coastal zone.

Finally, at several stations, a decrease in the $^{226}$Ra activities is observed near the seafloor (stations 13, 21, 44, 60, 64 and 77; Fig.7). Similar decreasing trends near the seafloor have been reported in the Southwest Atlantic and North Pacific for $^{230}$Th (Deng et al., 2014; Okubo et al., 2012), a reactive element that strongly adsorbs onto suspended particles. This trend for $^{230}$Th was explained by nuclide scavenging at the seafloor (Deng et al., 2014; Okubo et al., 2012). Radium-226—although it is much less particle-reactive than $^{230}$Th—and Ba may also be scavenged by resuspended particles near the seafloor and may adsorb onto the surfaces of Mn oxides (Moore and Reid, 1973). High particulate trace element concentrations were also observed at stations 26, 38, and 69 and may be related to nepheloid layers that impact the deep water column up to 200-300 m above the seafloor (Gourain et al., 2017 ; Menzel et al., 2017 ; this issue).

## 4.4. Estimation of $^{226}$Ra and Ba input fluxes into the West European Basin

A strong $^{226}$Ra positive anomaly is observed in the NEADWl between stations 1 and 21 and below 3500 m. On average, it is 3.3 dpm 100 L$^{-1}$ over this vertical section. This anomaly reflects a concentration difference between the $^{226}$Ra measured along GA01 and the $^{226}$Ra due to water mass mixing. This concentration difference is associated to the northward transport of the NEADWl, estimated to be $0.9 \pm 0.3$ Sv ($10^6$ m$^3$ s$^{-1}$) at 45 °N (GA01 section) (Daniault et al., 2016; McCartney,

1992). Therefore, the positive concentration anomaly can be converted into a $^{226}$Ra flux that has to be
added to this water mass, $F_{Input-Ra}$, calculated as follows:

$$F_{Input-Ra} = A \times T_{NEADWl} \ (1)$$

where $A$ is the mean positive anomaly of $^{226}$Ra (in dpm m$^{-3}$) determined by the OMP analysis;
$T_{NEADWl}$ is the transport associated with the NEADWl (in m$^3$ s$^{-1}$).
This $^{226}$Ra input may be associated with a sedimentary source and/or may result from the
dissolution of suspended particles. In a first place, we will study the hypothesis of the sedimentary
source; the suspended particle source will be discussed in a second place.
The NEADWl at 45 °N is made of up to 92 % of the 11 °N NEADWl endmember. Therefore, the
sedimentary input along the northward transport of the NEADWl is calculated across a sediment area
between 11 °N and 45 °N (Fig.S3). We consider the distance of 4209 km between 11 °N and the GA01
section and the distance of 1475 km between stations 1 and 21. This leads to a horizontal area of 6.21
$10^6$ km$^2$ (assuming a constant bathymetry), across which the sedimentary input is assumed to take place.
The $^{226}$Ra flux diffusing out of bottom sediments, $F_{Sed-Ra}$ (in dpm cm$^{-2}$ y$^{-1}$) can be calculated using
Eq. (2), assuming that the anomaly is entirely explained by the sediment source:

$$F_{Sed-Ra} = \frac{F_{Input-Ra}}{S} \ (2)$$

where $S$ is the surface area described above (in cm$^2$) and $F_{Input-Ra}$ is 1.67 $10^8$ dpm s$^{-1}$.
The calculated $F_{Sed-Ra}$ is 14.8 ± 3.1 $10^{-3}$ dpm cm$^{-2}$ y$^{-1}$, which is within the range of fluxes
reported in the literature. For example, Cochran (1980) reported $F_{Sed-Ra}$ in the range of
1.5 $10^{-3}$ dpm cm$^{-2}$ y$^{-1}$ for the Atlantic Ocean to 2.1 $10^{-1}$ dpm cm$^{-2}$ y$^{-1}$ in the Pacific Ocean (Fig.8). Li et
al. (1973) estimated $^{226}$Ra fluxes diffusing out of the sediment in the Southern Ocean and on the
Antarctic shelf of 6.2 $10^{-4}$ dpm cm$^{-2}$ y$^{-1}$ and 1.6 $10^{-3 \ dpm}$ cm$^{-2}$ y$^{-1}$, respectively. The $F_{Sed-Ra}$ calculated here
is thus slightly higher than the $^{226}$Ra sedimentary fluxes reported in the Atlantic Ocean by Cochran
(1980). Note, however, that the $^{226}$Ra fluxes released from the sediments vary locally as a function of
the sedimentary $^{230}$Th activity, bioturbation rates, but also as a function of the sediment type and
composition (Chung, 1976; Cochran, 1980). The $^{226}$Ra fluxes reported in the Atlantic Ocean by Cochran
(1980), which are the lowest of all basins, are only available for calcareous ooze type sediment
(Cochran, 1980). The NEADWl may cross different types of sediments along its northward path in the
Atlantic Ocean. This includes calcareous oozes, fine-grained calcareous sediments and clay (Dutkiewicz
et al., 2015). In particular, $^{226}$Ra diffusion is expected to be higher in these two latter sediment types
(Cochran, 1980).
As for Ba is concerned, the mean positive anomalies deduced from the OMP analysis is
7.0 nmol L$^{-1}$ leading to a $F_{Input-Ba}$ of 69.5 mol s$^{-1}$. In the same way as $^{226}$Ra, a Ba sedimentary flux $F_{Sed-Ba}$

of $3.19 \pm 1.4$ nmol cm$^{-2}$ y$^{-1}$ would be required to explain the Ba anomalies observed in the West European Basin. This flux is on the low end of the Ba sedimentary fluxes reported in different ocean basins, which range from 1.0 to 30 nmol cm$^2$ y$^{-1}$ (Chan et al., 1977; Falkner et al., 1993; McManus et al., 1999; Paytan and Kastner, 1996).

Alternatively, the dissolution of settling particles could also contribute to the $^{226}$Ra and Ba anomalies observed in the deep waters of the West European Basin. Assuming steady state, we may undertake a mass balance calculation for particulate $^{226}$Ra and Ba in the same box as described above (i.e., box defined between 11 °N and the GA01 section, between stations 1 and 21 –1475 km–and between 3500 m depth and the seafloor; Fig.S5). Particles enter the box from above as settling particles, but also horizontally, carried within the water masses at 11 °N that travel northward. Particles leave the box by different processes (accumulation in the sediment or northward transport by the water masses) or alternatively may dissolve while settling in the box. In the absence of precise information about the particulate $^{226}$Ra and Ba fluxes entering and exiting the box horizontally (i.e. the particulate $^{226}$Ra and Ba concentrations at 11 °N and at the GA01 section), we assume that they are of equal importance and therefore that they cancel each other in the mass balance calculation.

The vertical particulate flux entering the box, from above, can be calculated as follows:

$$F_{Part-x} = Cp_{3500} \times Vs \times S \ (3)$$

where $x$ is either $^{226}$Ra or Ba, $Cp3500$ is either the particulate $^{226}$Ra activities or the particulate Ba concentrations at 3500 m, $Vs$ is the settling speed for suspended particles and $S$ is the horizontal surface area described above (6.21 10$^6$ km$^2$).

We use the value of 0.007 dpm 100 L$^{-1}$ for the mean $^{226}$Ra particulate activity at 3500 m, a value that was reported for the Atlantic Ocean, Sargasso Sea (van Beek et al., 2007) and the value of 0.087 nmol L$^{-1}$ for the mean Ba particulate concentration at 3500 m, a value that was determined along the GA01 section (Lemaitre et al., this issue). We use the settling speeds (Vs) reported for suspended particles in the literature and that typically range from 100 to 1000 m y$^{-1}$ (Bacon and Anderson, 1982; Krishnaswami et al., 1976; Roy-Barman et al., 2002). The $F_{Part}$ thus obtained range from 1.4 10$^6$ to 13.8 10$^6$ dpm s$^{-1}$ for $^{226}$Ra, while the $F_{Part}$ range from 1.7 and 17.2 mol s$^{-1}$ for Ba. Of this total F$_{Part}$, some fraction may dissolve while settling, while the remainder will accumulate in the sediment. This dissolution flux is named F$_{dissolution-x}$, where x is either $^{226}$Ra or Ba. We use the sediment Ba accumulation rates reported by Gingele and Dahmke (1994) in the Atlantic Ocean to calculate the particulate Ba flux that exits the box (F$_{Accumulation-Ba:}$ 2.0 to 13.4 mol s$^{-1}$); hence, by difference the F$_{dissolution-Ba}$ is 0-15.2 mol s$^{-1}$ (Fig.10). This value is of the same order of magnitude of the $F_{Input-Ba}$

needed to explain the Ba anomalies (6.28 mol s$^{-1}$). Therefore, in the case of Ba, the dissolution of
settling particles may entirely explain the OMPA-derived anomalies. The sediment $^{226}$Ra accumulation
rates can be calculated from the Ba accumulation rates estimated above, using the $^{226}$Ra/Ba ratio
determined in sinking particles collected in the Sargasso Sea near the seafloor (i.e., 1.5 dpm µmol$^{-1}$; van
Beek et al., 2007). The sediment $^{226}$Ra accumulation flux thus calculated, $F_{Accumulation-Ra}$, is 2.9 10$^6$-
19.6 10$^6$ dpm s$^{-1}$, leading to $F_{dissolution-Ra}$ of 0-10.9 10$^6$ dpm s$^{-1}$ (Fig.10). Therefore, $F_{dissolution-Ra}$ cannot
account for more than 37 % of the required $^{226}$Ra flux ($F_{Input-Ra}$). This implies that even if the settling
speed is high (1000 m y$^{-1}$; high turnover of the particles), the particle dissolution cannot account for the
entire $F_{Input-Ra}$. The remaining (minimum of 63 %) therefore has to be sustained by $^{226}$Ra diffusion from
the sediments.
While the above calculations have to be taken with caution given the numerous assumptions in the
mass balance model, overall they suggest that the $^{226}$Ra positive anomalies observed in the West
European Basin may be explained entirely by $^{226}$Ra that diffuses out of the sediments. However, it
cannot be excluded that the dissolution of settling particles also contributes to the $^{226}$Ra enrichment. In
contrast, the Ba positive anomalies may be explained either by the diffusion of Ba from sediment or by
the dissolution of settling particles or by a combination of these two processes. These conclusions are in
line with the current knowledge about $^{226}$Ra and Ba sources in the deep ocean (Broecker et al., 1970;
Chan et al., 1976, 1977; Ku et al., 1980).

## 19  5. Conclusion

We investigated the distribution of dissolved $^{226}$Ra activities and Ba concentrations in the North
Atlantic Ocean along the GA01 section. To a first order, the $^{226}$Ra and Ba patterns appear to be
correlated to the water masses (e.g. high $^{226}$Ra and Ba in NEADWl in the West European Basin; low
$^{226}$Ra and Ba in Central Waters; slight increase of $^{226}$Ra in the MW). Using a mixing model, we show
that the measured $^{226}$Ra and Ba concentrations can be explained by conservative mixing for 58 % and
65 % of the samples respectively, notably at intermediate depth (mostly between 1000 m and 2000 m)
and slightly deeper in the middle of deep basins, away from the ocean interfaces. These locations
correspond to the depths where the waters are away from the main sources of $^{226}$Ra and Ba. $^{226}$Ra and
Ba can thus be considered as conservative tracers of water mass transport in the ocean interior at the
space scales considered here, namely, of the order of few thousand km.
Our study also highlighted several regions where significant input or loss of $^{226}$Ra and Ba takes
place. In the West European Basin, the deep waters (NEADWl) accumulate both $^{226}$Ra and Ba while
flowing from 11 °N to the GA01 section. Mass balance calculations suggest that these enrichments are
predominantly explained by sediment diffusion for $^{226}$Ra, with a possible contribution from suspended

particle dissolution, while both the sediment and suspended particle dissolution could significantly contribute to the Ba enrichments. This pattern contrasts with that observed in the deep waters collected elsewhere along the section that do not display Ba enrichments associated to the $^{226}$Ra enrichments. Bottom nepheloid layers may also contribute to the release of $^{226}$Ra (and Ba) to the deep water column at several stations. Interestingly, nepheloid layer processes seem to also act as potential removal of $^{226}$Ra (and Ba) at several other stations. Significant input of Ba—likely associated with meteoritic waters—is found in the upper water column near Greenland. Finally, $^{226}$Ra and Ba are removed from the upper water column, likely primarily due to biological mediated processes such as incorporation of $^{226}$Ra and Ba into barite ($BaSO_4$) that are presumably formed following the decay of settling organic matter and/or adsorption onto diatom frustules, a mechanism that would explain the $^{226}$Ra-Ba relationship reported here. Similarly, strong correlations were also found between Ba-Si and $^{226}$Ra-Si although no obvious process links $^{226}$Ra or Ba with Si, except maybe for the adsorption of Ba and ($^{226}$Ra) onto diatom frustules. It cannot be excluded, however, that the observed Ba-Si and $^{226}$Ra-Si relationships may result from the spatial coherence of different carriers overprinted by hydrodynamics.

Our study also provides evidence of significant decoupling between $^{226}$Ra and Ba. In the upper 200 m, the $^{226}$Ra/Ba ratios reach low values (<2 dpm $\mu mol^{-1}$), a pattern that has been observed in other regions and was related to acantharian skeletons that incorporate $^{226}$Ra preferentially to Ba (van Beek et al., 2007; Bernstein et al., 1998). Finally, deviations from the mean GEOSECS $^{226}$Ra/Ba ratios were observed in the shallow coastal waters of Greenland and Newfoundland: the predominant input of Ba due to input of meteoritic waters leads to lower $^{226}$Ra/Ba ratios whereas near the seafloor, the input of sedimentary $^{226}$Ra leads to higher $^{226}$Ra/Ba ratios.

The absence of a stable isotope for radium led geochemists to consider Ba as a stable analog for $^{226}$Ra because $^{226}$Ra and Ba display a similar chemical behavior, with the aim of using the $^{226}$Ra/Ba ratio as a chronometer for the thermohaline circulation. This study confirms that $^{226}$Ra and Ba behave similarly in the ocean interior away from external sources, both elements being predominantly conservative in the studied area over distances of the order of a few thousands of km. However, this study also highlights regions where $^{226}$Ra and Ba deviate from a conservative behavior, an important consideration when considering the balance between the large-scale oceanic circulation and biological activity over long time scales. Decoupling between $^{226}$Ra and Ba has been observed, in most cases at the ocean boundaries as the result of dissolved $^{226}$Ra and Ba external sources. In addition, suspended particle dissolution may differently impact the dissolved $^{226}$Ra and Ba content of intermediate and deep waters (as shown for the NEADWl); such process would therefore potentially modify their $^{226}$Ra/Ba ratios and would complicate the use of this ratio as a chronometer. Inclusion of the different sources and sinks and particle/dissolved interactions in global ocean models should help to refine the use of the

$^{226}$Ra/Ba ratio as a clock to chronometer the thermohaline circulation, as was proposed several decades
ago during the GEOSECS program.

# Figure Caption

**Figure 1:** Station locations of the GA01 section between Portugal and Newfoundland in the North Atlantic (black and blue dots). Stations
investigated for $^{226}$Ra and Ba are marked as blue dots. The main currents and water masses in the North Atlantic are also represented. The
major hydrothermal vents located near the GA01 section are indicated by black triangles. Stations investigated during the US-
GEOTRACES-GA03 section, also conducted in the Atlantic Ocean, are reported on the lower panel (red dots).
**Figure 2:** Potential temperature-salinity diagram—including a zoom for bottom waters—of the water samples (colored dots) from the
GA01 section. The properties of the source water types (based on García-Ibáñez et al., 2017; This issue) used in the Optimum
Multiparameter (OMP) analysis are reported with white circles. Isopycnals are also plotted (potential density referenced to 0 dbar).
**Figure 3:** Distribution of salinity (CTD data) along the GA01 section. The different water masses are also reported, following García-
Ibáñez et al. (2017; this issue). The station numbers are found on top of the figure. The sampling depths for $^{226}$Ra are shown for each
vertical profile (black dots).
**Figure 4:** Distribution of (a) dissolved $^{226}$Ra activities (dpm 100 L$^{-1}$), (b) dissolved Ba concentrations (nmol L$^{-1}$) and $^{226}$Ra/Ba ratio
(dpm µmol$^{-1}$) along the GA01 section. Station numbers are found on top of the panels. The sampling depths are shown for each vertical
profile (black dots).
**Figure 5:** Relationships between $^{226}$Ra and Ba (red dots) and between $^{226}$Ra and Si(OH)$_4$ (blue dots) along the GA01 section in the North
Atlantic. The best linear fit for the two plots is also reported (R=0.93 for the two plots). The slopes of the relationships between $^{226}$Ra-Ba
and between $^{226}$Ra-Si(OH)$_4$ are expressed in $10^{-2}$ dpm nmol$^{-1}$ and in $10^{-2}$ dpm µmol$^{-1}$, respectively.
**Figure 6:** Difference between the measured concentrations and those calculated by the OMP analysis, for $^{226}$Ra (a), Ba (b) and (c)
$^{226}$Ra/Ba ratio along the GA01 section. Positive anomalies reflect recent tracer addition, while negative ones reflect recent tracer removal.
Station numbers are found on top of the panels.
**Figure 7:** Vertical profiles of dissolved $^{226}$Ra activities and Ba concentrations determined along the GA01 section: (a) West European
Basin; (b) Iceland Basin and Irminger Basin, (c) the Greenland and Newfoundland margins, and (d) Labrador Basin. As a comparison, the
conservative $^{226}$Ra and Ba vertical profiles derived from the OMP analysis are also reported in solid grey lines. The discrepancy between
the two vertical profiles indicates deviation from the conservative behavior and reflects either an input of $^{226}$Ra or Ba (positive anomalies
highlighted in red; same color code as Fig.6) or a removal of $^{226}$Ra or Ba (negative anomalies highlighted in blue; same color code as
Fig.6). The OMP analysis has not been solved for the shallow coastal stations 53 and 78. The $^{226}$Ra/Ba ratios are also reported, together
with the mean GEOSECS $^{226}$Ra/Ba ratio (2.2 ± 0.2 dpm µmol$^{-1}$; black dashed line) together with its one standard deviation (grey shaded
areas). Note that the scale may be different from one station to the other and the vertical axis was cut to 1000 m. The seafloor is
represented by the bottom axis.
**Figure 8:** Geographical variation of $^{226}$Ra activities (red dots), salinity (blue dots), temperature (yellow dots) and Si(OH)$_4$ concentrations
(green dots) in AABW and the NEADWl between 60 °S and 45 °N (GA01 section) in the Atlantic Ocean based on data from the
GEOSECS and TTO programs. The $^{226}$Ra activities, salinity, temperature and Si(OH)$_4$ concentrations from GA01 are represented with
open circles. The values used as endmembers for the OMP analysis are also identified by the black circles. The shaded area represents the
region where transformation of the AABW into NEADWl takes place.
**Figure 9:** $^{226}$Ra fluxes diffusing out of the sediment in relationship with bottom water $^{226}$Ra activities determined in different oceanic
basins (P=Pacific Ocean, A=Atlantic Ocean, I =Indian Ocean and AA=Southern Ocean) by Cochran (1980). The $^{226}$Ra flux calculated in
this study to explain the positive anomalies in the West European Basin is also reported (red dot).
**Figure 10:** $^{226}$Ra and Ba flux estimations: $F_{Part\text{-}x}$ is the particulate flux entering the box and $F_{Accumulation\text{-}x}$ is the sediment accumulation flux
and $F_{Dissolution\text{-}x}$ is the flux of particle dissolution assuming that all the settling particles dissolve. $x$ is either $^{226}$Ra or Ba. Both maximum and
minimum values are shown for $F_{Part\text{-}x}$ and $F_{Accumulation\text{-}x}$. $Max\ F_{Dissolution\text{-}x}$ represents a maximum value since it is calculated by subtracting
the minimum value of $F_{Accumulation\text{-}x}$ from the maximum value of $F_{Part\text{-}x}$. $F_{Dissolution\text{-}x}$ is thus comprised between zero (if all $F_{part\text{-}x}$ accumulates
in the sediment) and this latter value.
**Figure S1:** Comparison of the vertical profiles of dissolved $^{226}$Ra at stations 1 and 13 of the GA01 section (black and red dots,
respectively) and station 1 of the GA03 section (U.S.-GEOTRACES; blue dots) off Portugal.
**Figure S2:** Vertical profiles of dissolved $^{226}$Ra activities and dissolved Ba concentrations with the conservative $^{226}$Ra and Ba vertical
profiles derived from the OMP analysis, $^{226}$Ra/Ba ratios, Si(OH)$_4$ concentrations, salinity (black line) and potential temperature (red line)
for (a) the Iberian margin and the West European Basin, (b) the Iceland Basin and the Irminger Sea, (c) the Greenland margin, and (d) the
Labrador Sea and the Newfoundland margin. Note that the scale may be different from one station to the other and the vertical axis was cut
to 1000 m. The bottom is represented by the bottom axis.
**Figure S3:** Location of each endmember source water types (SWTs) used for the OMP analysis (black circles). The surface of the basin,
$S$, used to calculate the fluxes is represented by the grey hatched area.
**Figure S4:** Satellite Chlorophyll-a concentrations (MODIS Aqua from http://oceancolor.gsfc.nasa.gov), in mg m$^{-3}$ during the GA01 cruise
in (a) May 2014 and (b) June 2014. The dashed line indicates the location of the GA01 section. Stations investigated in this work are
indicated by dots. White dots indicate the stations investigated during the corresponding month.
**Figure S5:** Schematic box model used to calculate the input fluxes in the West European Basin: $F_{Sed-x}$ is the flux diffusing out of bottom
sediments, $F_{Part\ -x}$ is the vertical flux of particles entering the box from above, $F_{Accumulation-x}$ is the flux of particles accumulating in the
sediment and $F_{H-In-x}$ and $F_{H-Out-x}$ represent horizontal fluxes of dissolved species or particles coming in and out of the box due to transport,
respectively. $x$ is either $^{226}$Ra or Ba.
**Table S1:** Characteristics and location of each endmember source water types (SWTs).
**Table S2:** $^{226}$Ra activities, Ba concentrations, $^{226}$Ra/Ba ratios, potential temperature and salinity at the different stations of the GA01
section.

## 1 Acknowledgement

The present research and Emilie Le Roy's fellowship are co-funded by the European Union and the
Région Occitanie-Pyrénées-Méditerranée (European Regional Development Fund). We are grateful to
the captain and crew of the N/O *Pourquoi Pas?*. The GEOVIDE project is co-funded by the French
national program LEFE/INSU (GEOVIDE), ANR Blanc (GEOVIDE, ANR-13-BS06-0014) and
RPDOC (ANR-12-PDOC-0025-01), LabEX MER (ANR-10-LABX-19) and IFREMER. The
GEOVIDE cruise would not have been achieved without the technical skills and commitment of
Catherine Kermabon, Olivier Ménage, Stéphane Leizour, Michel Hamon, Philippe Le Bot, Emmanuel
de Saint-Léger and Fabien Pérault. We are grateful to Manon Le Goff, Emilie Grosstefan,
Morgane Gallinari and Paul Tréguer for $Si(OH)_4$ sampling and analysis. This work was also co-funded
by the French national program LEFE/INSU "REPAP" (PI S. Jacquet) and the U.S. National Science
Foundation (PI M. Charette, OCE- 1458305; OCE- 1232669). For this work M.I. García-Ibáñez and
F.F. Pérez were supported by the Spanish Ministry of Economy and Competitiveness through the
BOCATS (CTM2013-41048-P) project co-funded by the Fondo Europeo de Desarrollo Regional 2014–
2020 (FEDER). Several figures were constructed using Ocean Data View (Schlitzer, 2003). Therefore,
R. Schlitzer is warmly thanked. Satellite chlorophyll-a visualizations used in this study were produced
with the Giovanni online data system, developed and maintained by the NASA GES DISC.

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

4

Figure 1

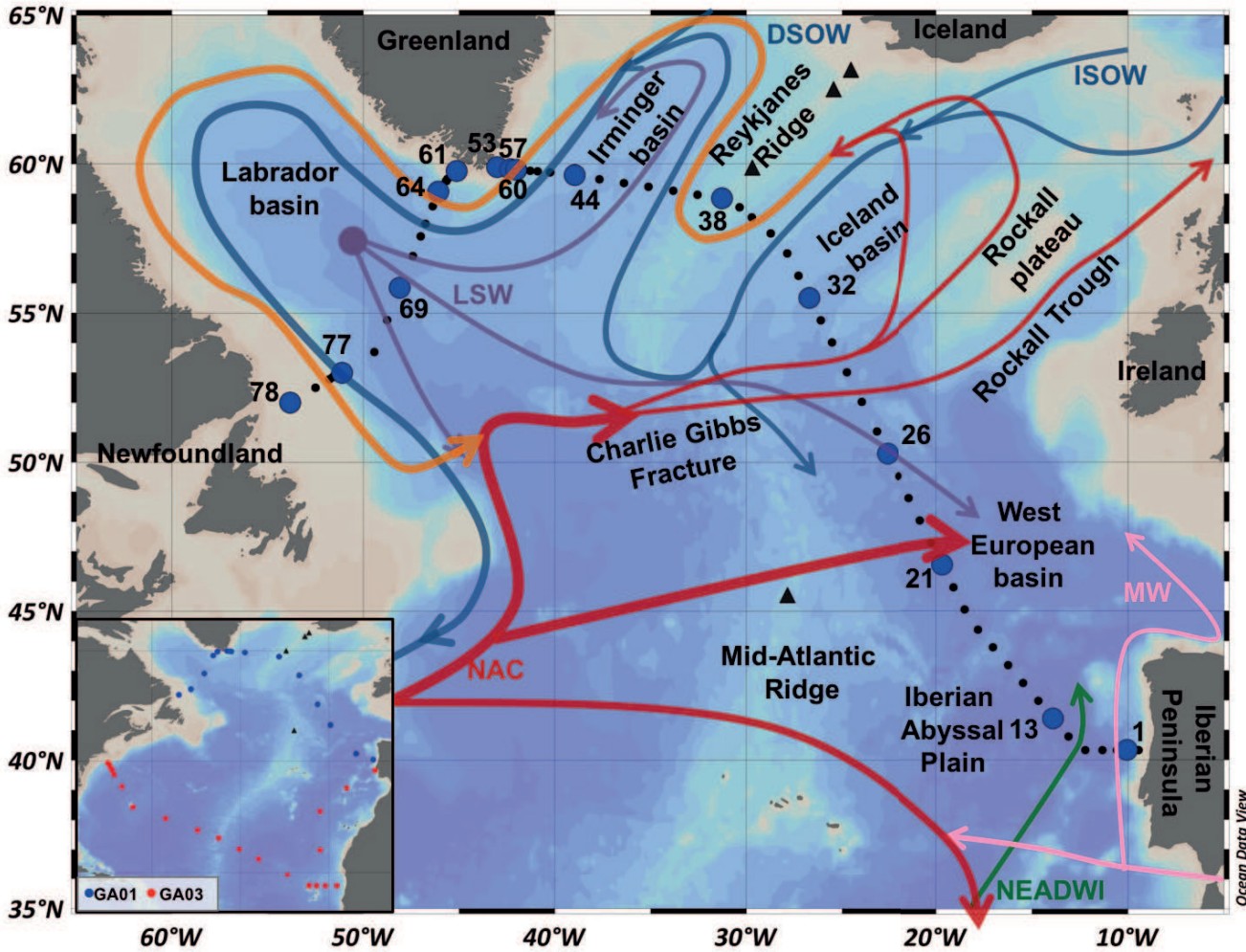

# Figure 2

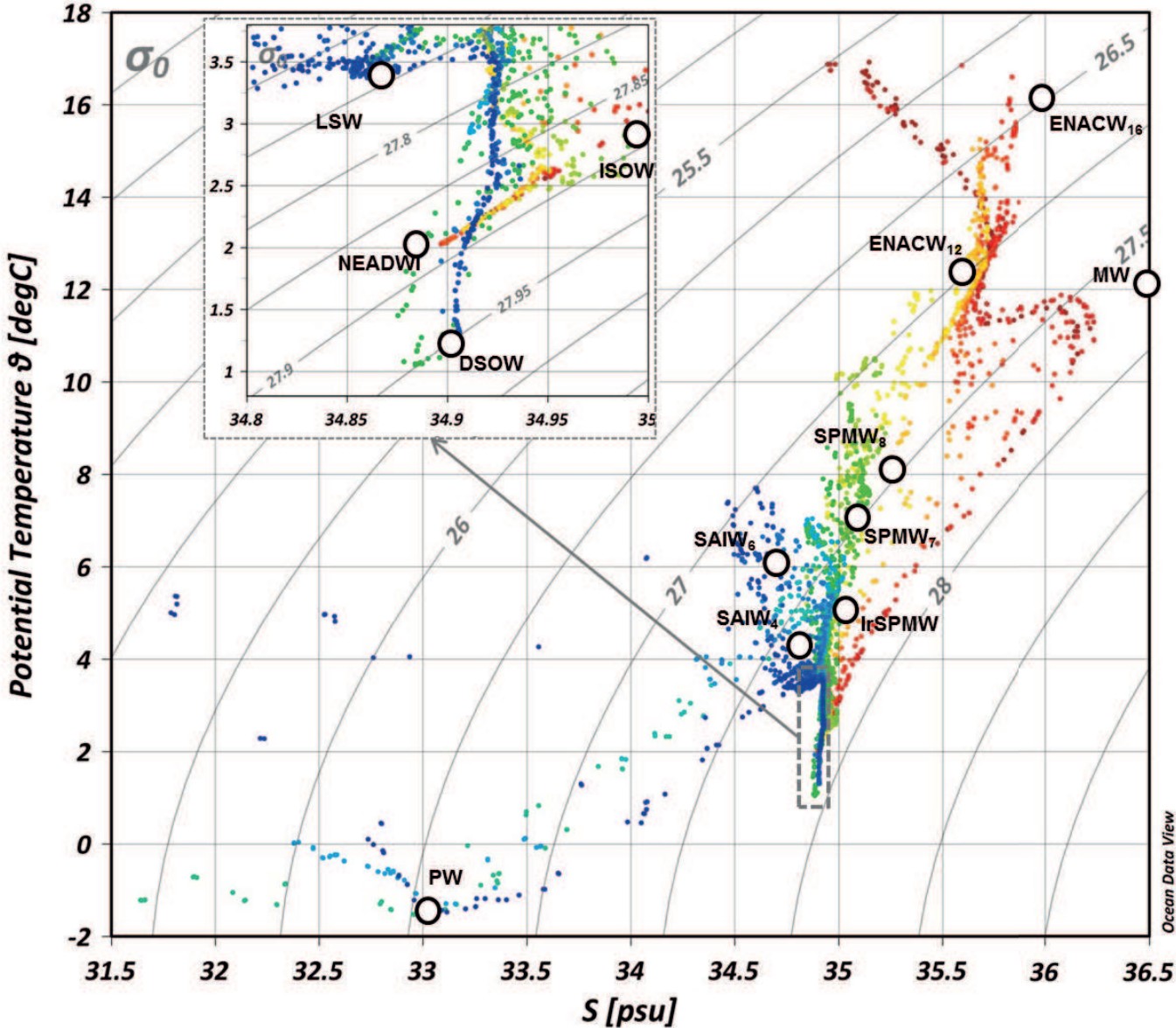

# Figure 3

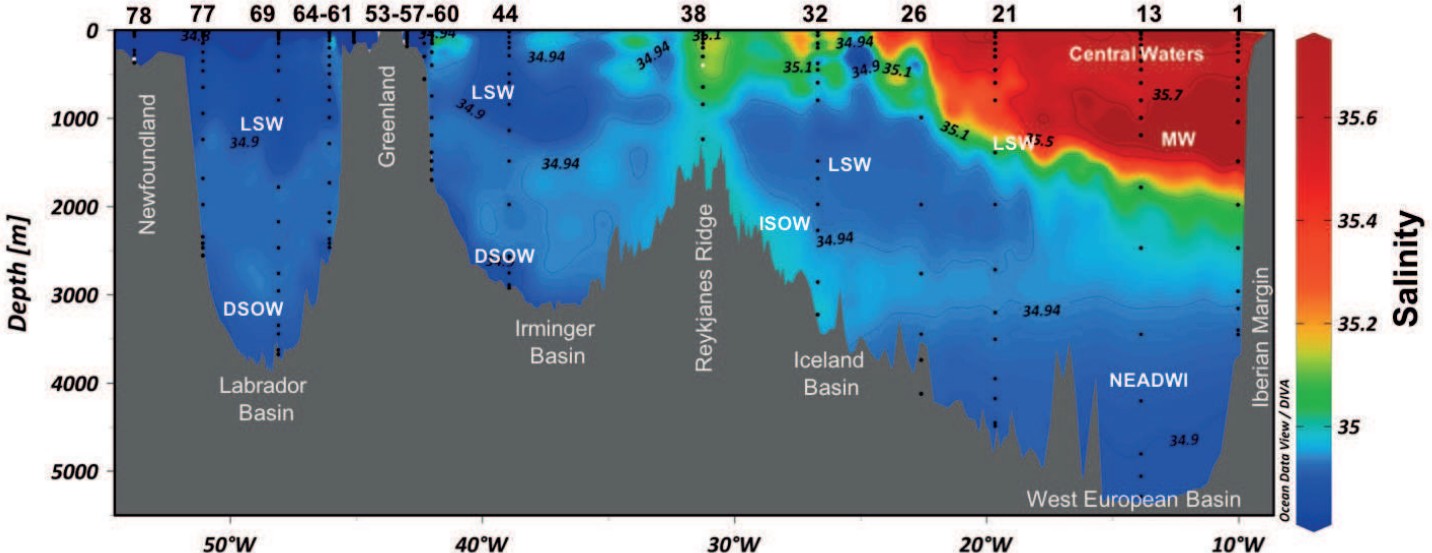

# Figure 4

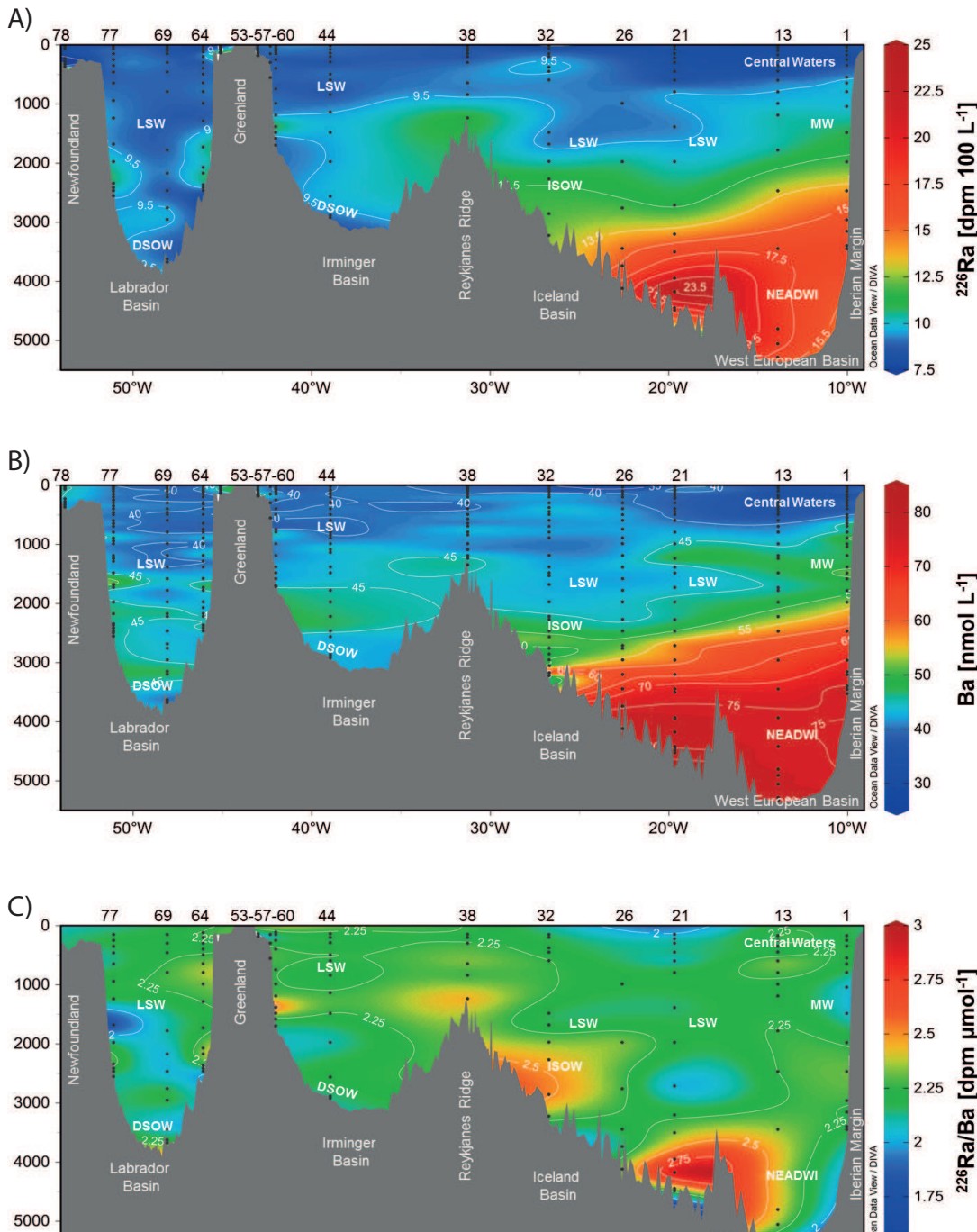

Figure 5

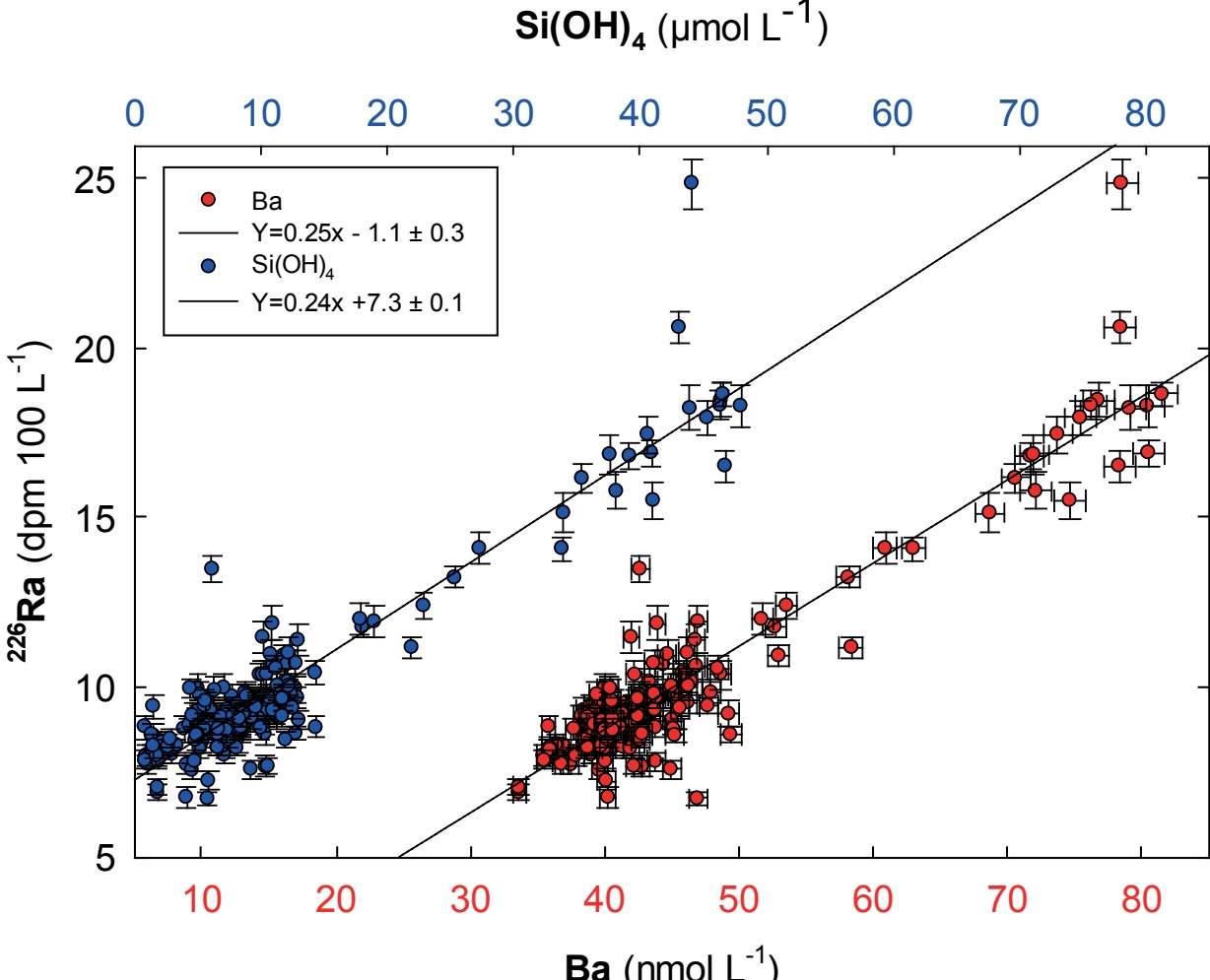

# Figure 6

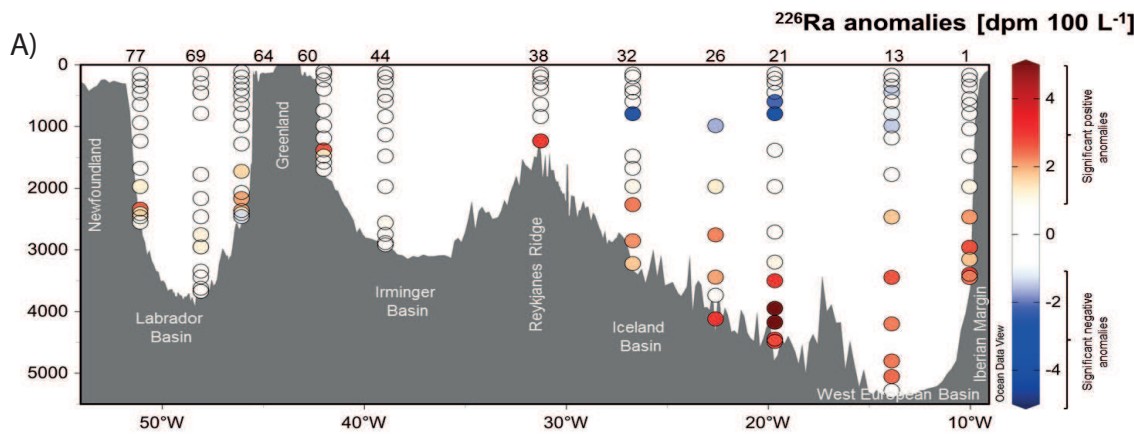

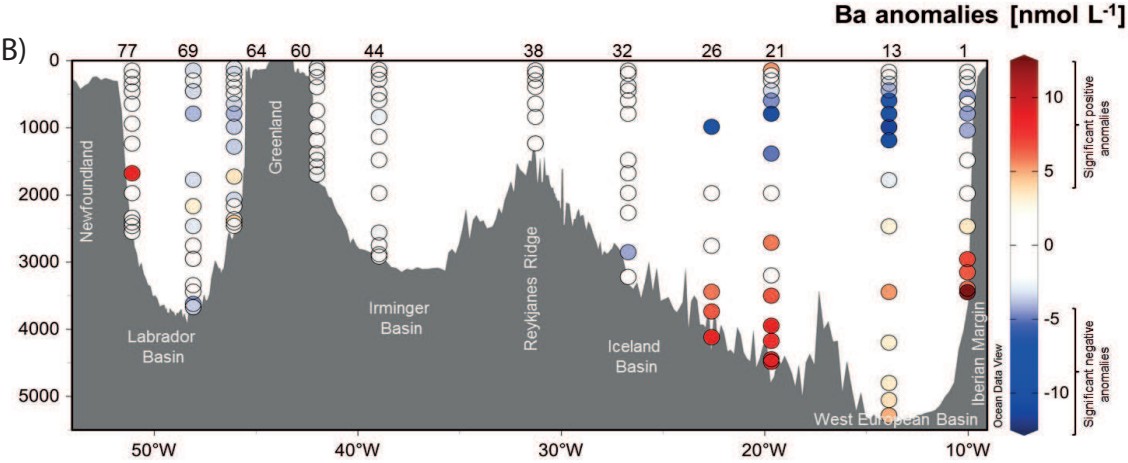

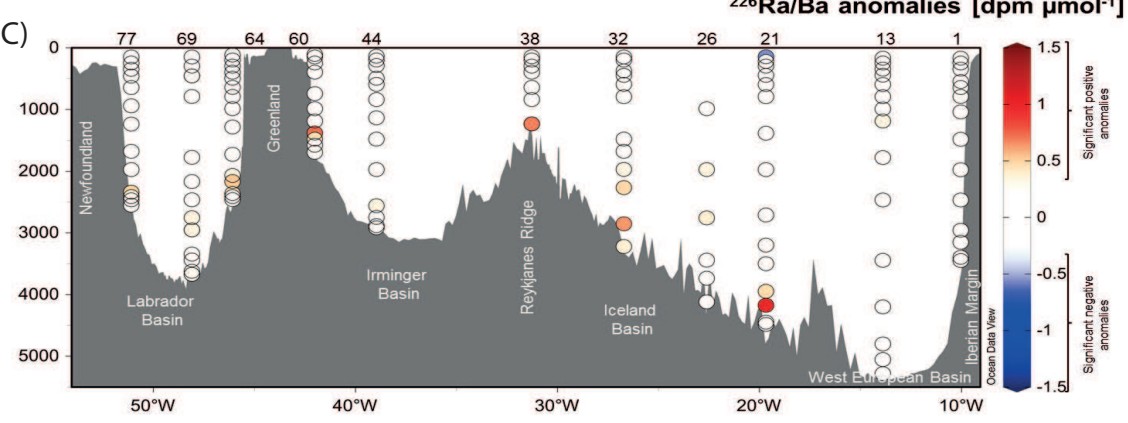

# Figure 7

## A) West European Basin

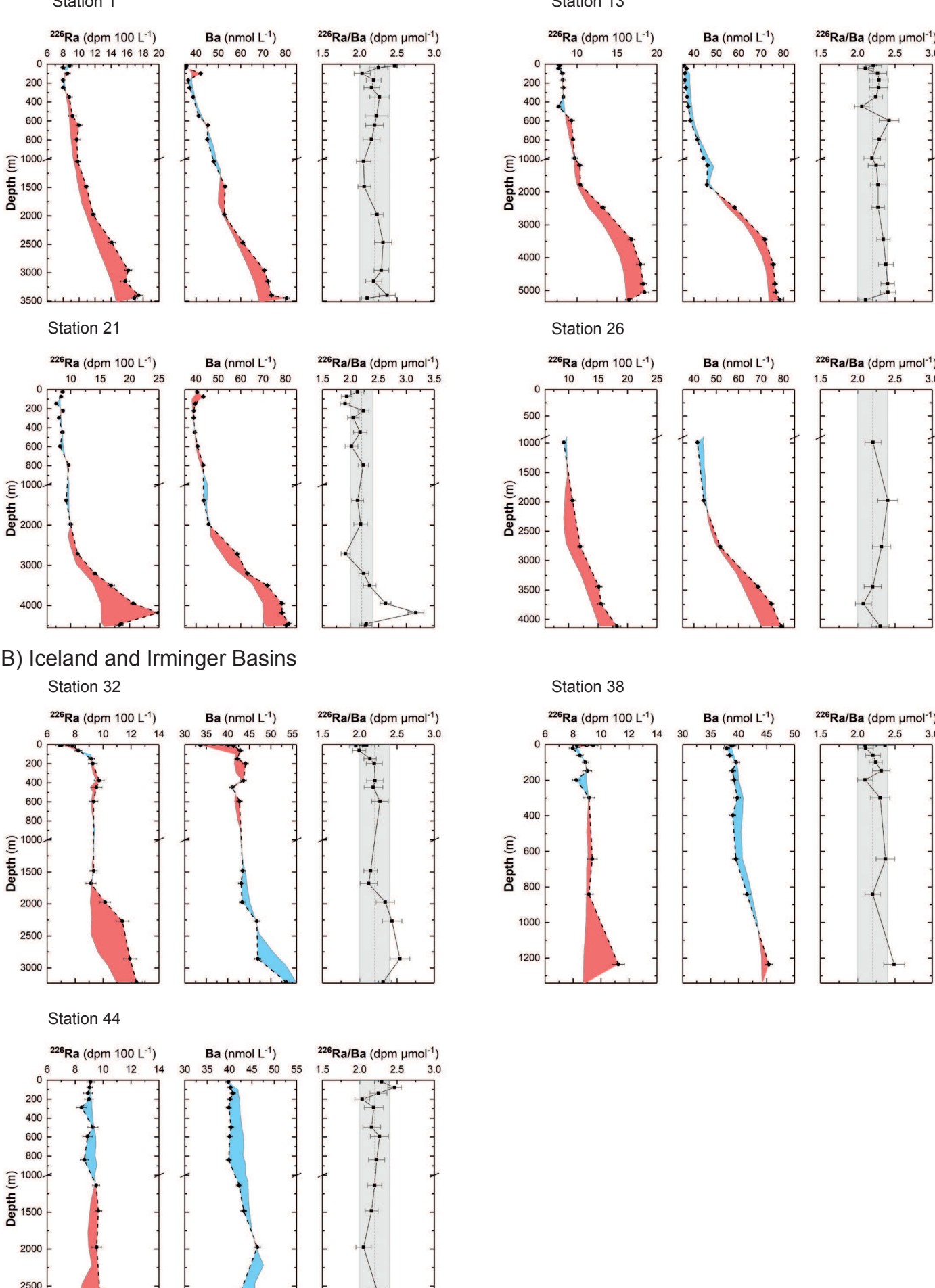

# Figure 7

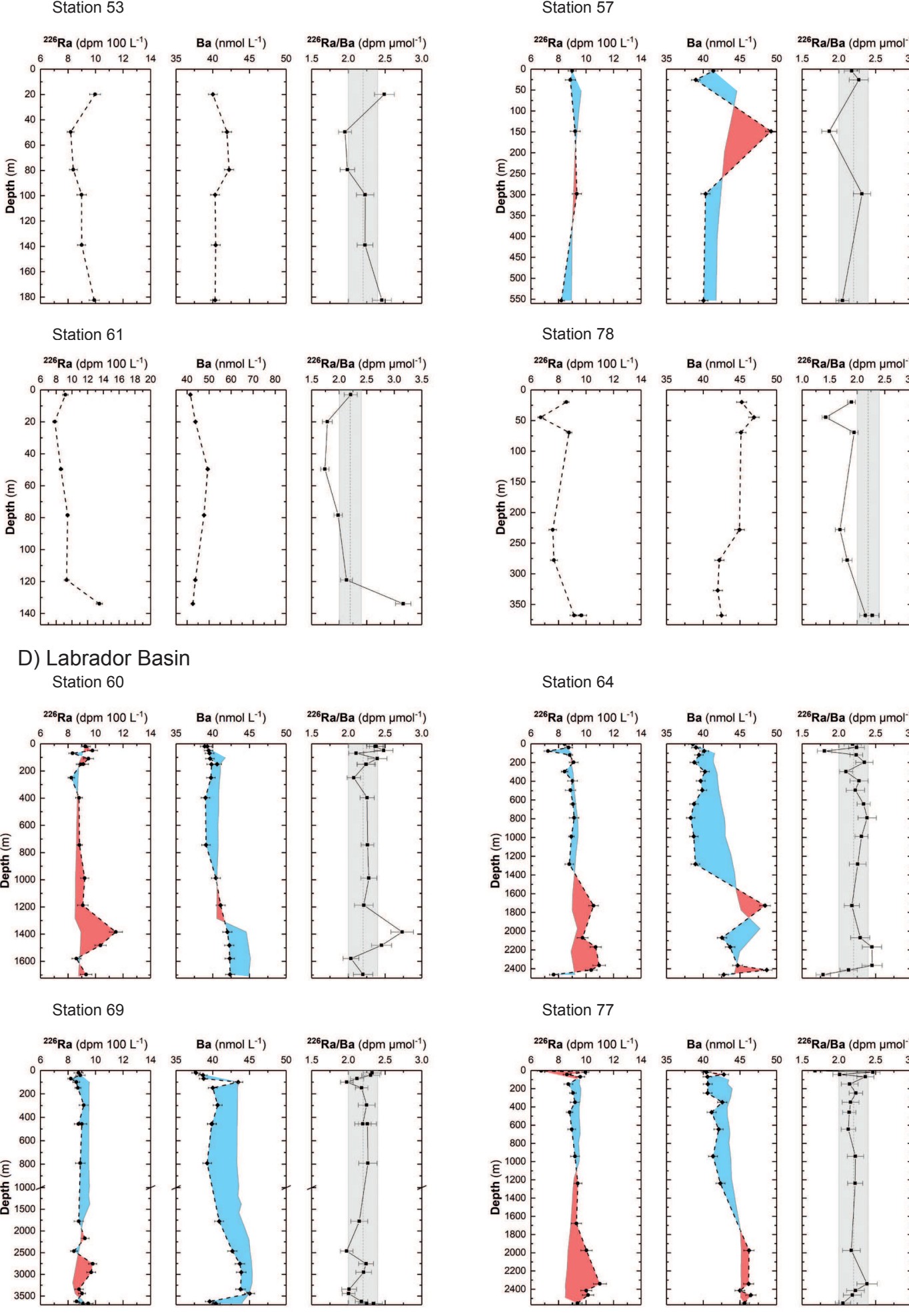

C) Greenland and Newfoundland Margins

D) Labrador Basin

# Figure 8

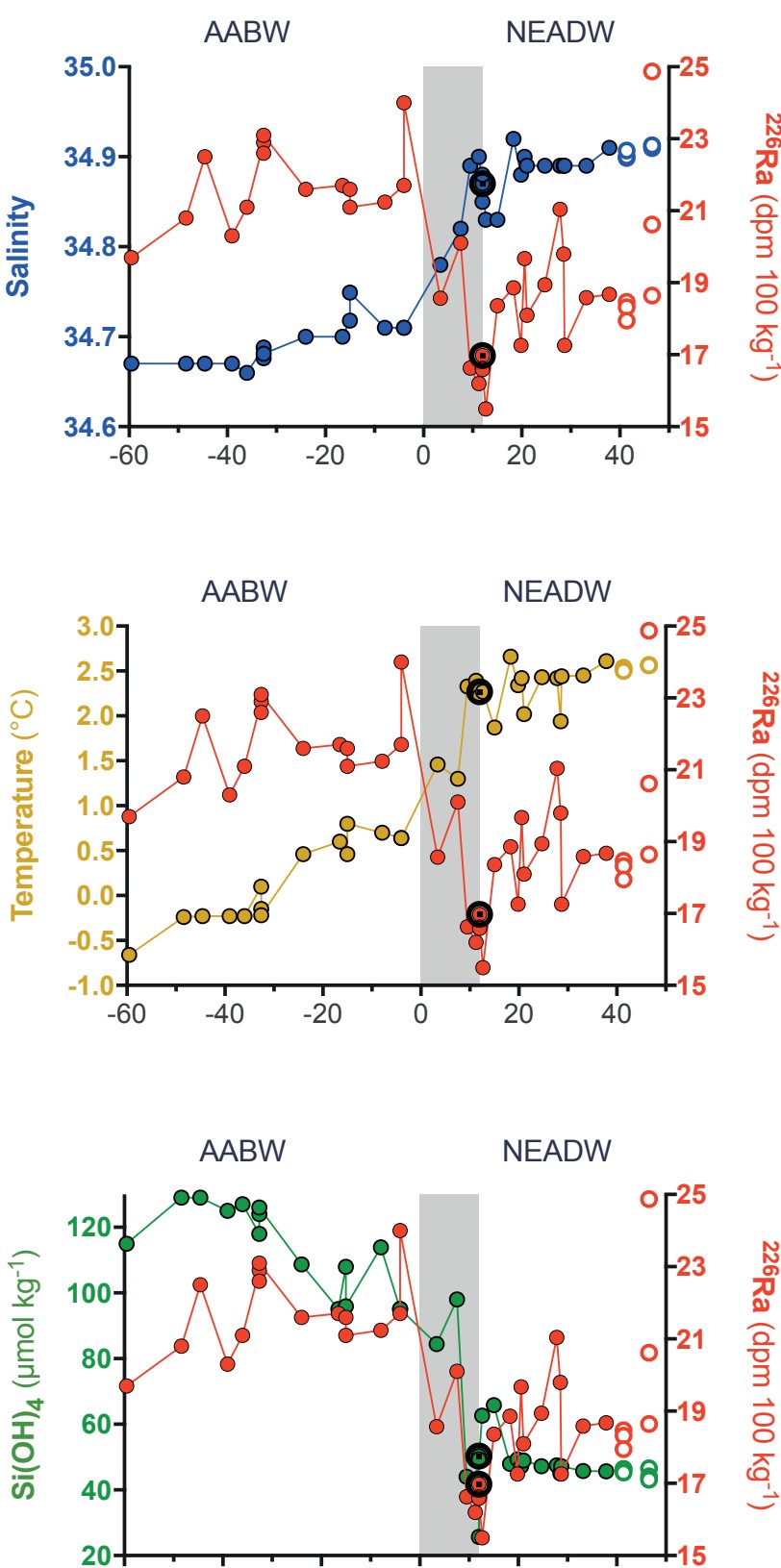

# Figure 9

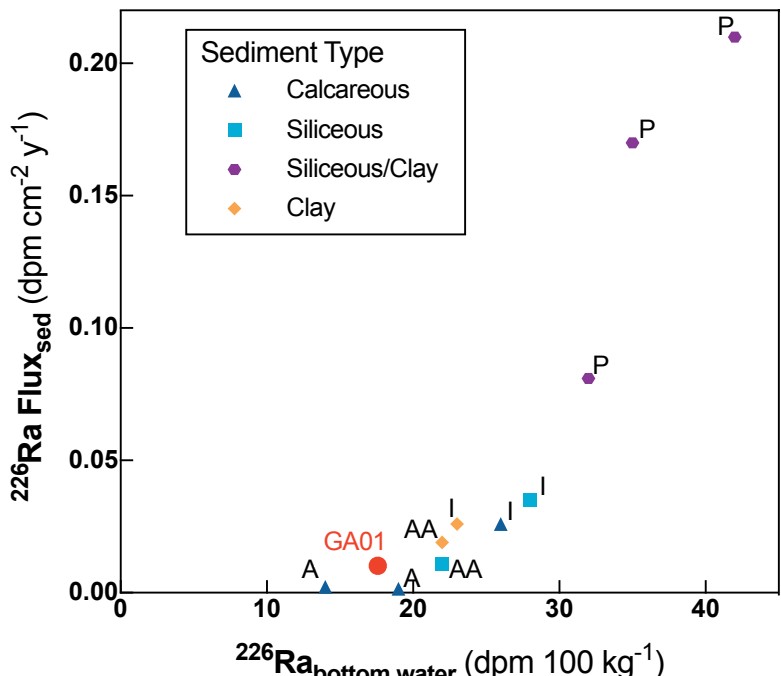

# Figure 10

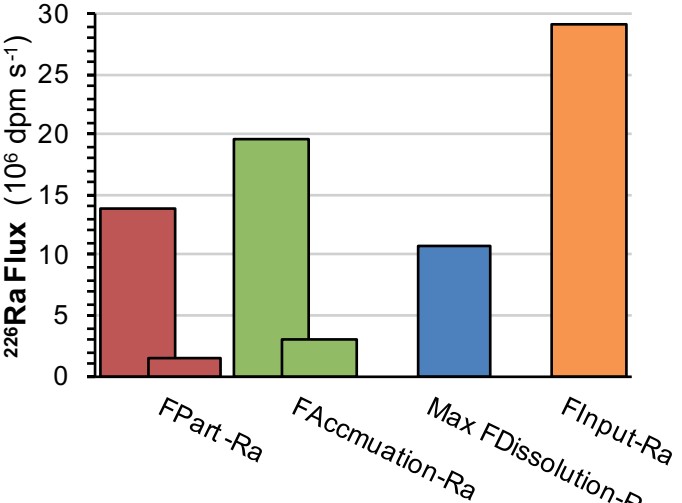

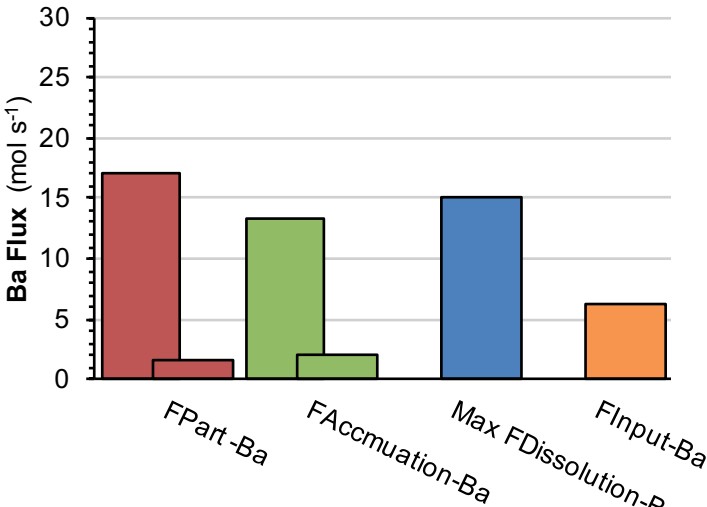