# Peer review of "The 226Ra-Ba relationship in the North Atlantic during"

_Biogeosciences, 2017_

## Referee Comment (RC1) · Anonymous Referee #1 · 23 Nov 2017

The authors present a study of seawater 226Ra and Ba in the North Atlantic along the GEOTRACES GA01 transect across regions that involve different water masses mixing as a crucial part of the global ocean circulation. The purpose of this paper is to improve our understanding of the relationship between 226Ra and Ba, two commonly used tracers in studying large-scale ocean circulation, and to investigate their conservative/non-conservative behaviours in the study areas. It has been a long-standing debate over the last few decades about the non-conservative behaviours of seawater 226Ra and Ba and how such behaviours restrict the use of 226Ra and Ba as conservative tracers in the large-scale ocean circulation. I think the paper has nicely approached these issues with their new dataset in the North Atlantic. The authors also use the OMP model to identify any anomalies associated with additional inputs or removals of 226Ra and

[Figure]

Ba in different regions, although I think that in most of the cases the anomalies cannot be precisely quantified or qualified. Nevertheless, such information is still valuable for the community and may help to improve our understanding of 226Ra and Ba cycling in the ocean circulation. I would recommend the manuscript for publication in this special issue in Biogeosciences Discussions. However, I list a few issues below, which I think the authors should consider in their revision.

1. Do seawater 226Ra and Ba behave similarly in the ocean?

The strong correlation between seawater 226Ra and Ba has been commonly used as a result of their similar chemical behaviour. Such an argument is perhaps not as convincing as it used to be since we have seen many pieces of evidence that 226Ra and Ba have different sources and sinks, and can behave differently in different regions (as discussed in Section 4.1 in the manuscript). This is a similar argument for Ba-Si; these two elements seem to be strongly correlated, despite the fact that they have such different chemical cycles in the ocean. Any two elements can be correlated to some extent. However, without fully understanding their removal or regeneration mechanisms, such correlation could also be a coincidence. I think the authors should make that clear in the manuscript, particularly in abstract, conclusion and Section 4.1.

2. OMP analysis

One thing that concerns me a little is the uncertainty and accuracy of the OMP analysis. In Section 4.2, the authors mentioned that the OMP analysis suggests that there are 41-66% of MW present between 1000-1600m at Station1 and 21, 60% of ISOW between 2700-3000 m in Station32, and 54% of ISOW between 2100 and 3000m in Station69 and 77. Both MW and ISOW are identified with a relatively low 226Ra/Ba ratio (1.7-1.8, Table S1), compared to 2.0-2.3 in other end-members. So in theory, I would expect to see low 226Ra/Ba in these locations. However, apart from Station69, most of the stations mentioned above either show no change in 226Ra/Ba or show an even higher ratio (i.e. Station32). It would be helpful to show the OMP estimate of

226Ra/Ba in Fig.8 as well.

I am also worried that if the estimate of OMP is correct, the absence of low 226Ra/Ba in the intermediate waters (e.g. Station21, 1000-1600m), where >50% was expected to be MW, would suggest that 226Ra/Ba is perhaps not as conservative as we thought even in the intermediate waters.

3. Removal of 226Ra and Ba

The authors suggest that phytoplankton blooms in the Labrador Sea may explain the negative anomalies of 226Ra and Ba in the areas. I think this statement requires more clarification and work. At least from the Ba point of view, we know that Ba removal (barite formation) is not a direct function of POC flux. In this case, the Labrador Sea (Station60, 64, and 69) shows nearly 10% Ba deficiency in water columns from 200 to 1500 m, according to the OMP analysis (Fig.8d). Such a huge deficiency for a relatively long-residence-time element is unlikely to be explained by seasonal phytoplankton blooms.

4. The uncertainty of estimated sediment 226Ra flux

In Section 4.4, equations (1) and (2) suggest that the uncertainty of sediment 226Ra flux comes from A (positive anomaly), T (transport rate) and S (surface area.) At the moment, a 20% error in the calculated 226Ra flux does not include the 33% error from the transport rate. I think the authors should consider the error propagation from each component in the flux calculation.

5. How much tolerance does seawater 226Ra/Ba need to be used as conservative tracers (clock) to chronometer the thermohaline circulation?

Despite that the authors suggest that 226Ra and Ba in intermediate waters are mostly conservative, the readers would not know how sensitive the seawater 226Ra/Ba is as a tracer to chronometer the thermohaline circulation. For example, based on our current understanding of the time for the global ocean circulation, how much decay of

226Ra/Ba is expected? Considering that there might still be some non-conservative behaviour (~30%) involved during the ocean basin scale mixing, how much tolerance does seawater 226Ra/Ba have if we want to use it for such an application? The authors could perhaps include some of this discussion in the paper.

Minor comments:

P3L33 R/V Pourquoi Pas?, the question mark seems to be a typo. P5L7-8 I understand that, in most of the cases, the particulate Ba is low (<1% of the total Ba). I think this needs to be checked for samples in nepheloid layers and hydrothermal plumes as well. P13L25-28, some references are required for this statement. P16L2, would not Ba be scavenged by Mn oxides as well? Fig.6. It would be much more clear if AABW and NEADWI are labeled on the figures to show the transformation of water properties. Fig.9. This figure shows some interesting observations. One thing worthy of notice is that the trend is not a linear relationship. I think that there are many reasons (sediment-water contact residence time, water age, and scavenging etc) to explain why sediment 226Ra flux and bottom water 226Ra concentration are not linearly correlated. This may be something interesting that the authors can point out to their readers in the discussion.

---

## Referee Comment (RC2) · Anonymous Referee #2 · 30 Nov 2017

General This manuscript brings new data concerning the distribution 226Ra activity and dissolved Ba in the North and Sub-Arctic Atlantic and the Labrador Sea. Authors compare the distributions of Ra and Ba with those of silicic acid, salinity and temperature. Using an OMP approach they estimate the relative importance of conservative mixing in setting the distributions of Ra and Ba. These appear highly correlated (despite differences in source terms) indicating water mass mixing as well as similar biogeochemical processes controlling their distribution. The potential of Ba as the stable analogue of 226Ra (as was noted as early as the 1950's and subsequently during the GEOSECS era) is underlined by the authors, opening a perspective of utilizing the 226Ra/Ba ratio for water mass dating. In general the paper is well organized and reads rather easily. Overall authors give interesting insights on the different possible controls

on 226Ra, Ba by source functions, internal biogeochemical processing and water mass mixing in this less studied northern Atlantic domain.

Specific comments P1 Lines 29-30: what about impact of radio-decay ?

P3 Line 5: explain parallel carriers

The description of the water masses and circulation as shown in Figure 1 could be more accurate: P 6, Line 24: from Fig. 1 it appears that stations 1 to 26 cover the section between the Iberian Peninsula and the Rockall Through, and not till Reykjanes Ridge as indicated in the text P 6, Lines 27-28: From Figure 1 splitting of the NAC occurs west of the Mid Atlantic ridge, not at the MAR as indicated in text. P6, Lines 30-31: the southern branch of the NAC appears to flow into the Iberian abyssal plain, rather than into the West European basin.

P9, Lines 17-18: positive intercept on Ba axis is explained as either resulting from larger river input relative to 226 Ra and larger 226Ra input relative to Ba at the sediment interface; it is not clear how the latter situation may lead to a positive intercept on the Ba axis.

P9: Lines 20-21: slope of regression curves 226Ra-Ba. Alert the reader that units in Fig 5 are not the same as those for the global ocean Ra/Ba ratio. P9 Line 25: change sentence "... small fractionation between Ra and Ba during these processes" to "indicates the fractionation between Ra and Ba during these processes is small".

P10 Lines 5 to 15: Some words of explanation are needed when discussing varying Ra to Si ratios between the major basins. A look at the Sarmiento et al. Nature paper of 2007 could be helpful.

P10 Line23: "These waters then sink and circulate northward into the Atlantic Ocean" specify that these 'waters' are mainly AAIW and also to some extend SAMW.

P10, Line 25 (Fig. 6) and onwards: What explains this abrupt decrease of 226Ra between Eq. and  $11^{\circ}N$ ? What is the rationale to consider these meridional profiles

as a continuum, or to what extent is this reconstructed meridional section of 226Ra and the other variables depending on the basin they originate from (i.e. west or east Atlantic Basin)? This needs to be clarified. Line 32: clarify the meaning of 'coincide'; you mean geographical coincidence or compositional similarity?

P10 Lines 34-35: sentence unclear. 226Ra activities are high when waters reach  $40^{\circ}$ N. So they are high compared to the western part of the GA01 section (not the eastern part) ..

P11 Line 13: presence of the MW at about 41% - 66%; % of what ? Specify.

P12 Lines 25- 35: What may cause the absence of a positive anomaly for Si, while the anomaly is strong for Ra ? Is it because the region is dominated rather by coccolithophorids (compared to the western part of the section, where the diatoms dominate), and so sediments may be relatively poor in Si ? Comment please.

P14: title of section 4.3.2 lacks a verb ?

P14 Line14: Acantharians are invoked to explain lower Ra/Ba ratios. Is there any direct evidence during the GEOVIDE cruise for the presence of these organisms ?

P14-15: Differentiation of west and east regarding dominance of diatoms (west) vs coccolithophorids (east) as such is not sufficient to explain the 226Ra distribution. Is it known that these two phytoplankton groups accumulate Ra and Ba in a different way? Please comment further.

P16 Line28: ... "which is within the range of fluxes" .. in fact the calculated flux is clearly larger than the reported fluxes in the literature, so stating that the statement is not appropriate.

P17 lines 15 - 20: It is not because Ra and Ba contents in particles are much lower than the contents in solution, that particle dissolution should be considered as minor. This depends on turn- over rate of particulate vs dissolved phase.

P17 Lines 31-32: Potential for 226Ra/Ba ratio to be used as a clock for THC. From the data here in the North Atlantic there is but poor evidence that with time 226Ra activity relative to Ba is decreasing. Also globally the Ra/Ba ratio appears to stick closely to the 2.2 value. Can you comment on what the effective perspective is for the use of this ratio as a chronometer ? Is it a question of insufficient precision on the present day data ? If so what needs to be done ? A few lines discussing the status of this issue would be welcome.

P18 Line 1-2: The absence of Ba enrichment in deep waters in the western part of the section, while Ra is enriched) is not really discussed. What could be the possible mechanisms?

Figures and tables: Legends should detail all acronyms (water masses) shown or listed.

Table S2: It would be useful to indicate the CTD number of the casts. Why not add also the Si(OH)4 profiles ? Giving depth values with a decimal seems not realistic given natural variabilities due to waves, swells .. Why are activity and concentration data not expressed per unit weight (Kg) rather than per unit volume, as is standard practice? Also for Ba, the indicated 1.5% precision implies the numbers should be rounded to the first decimal. Some numbers may be suspicious: examples are station 21: 226Ra = 24.87 at 4176m; temperature at 794m; station 32: 226Ra at 794m; please check. Station 60: Why is the depth range not continuous ? Are the data from two different CTD casts ? This is what Sal and Temp suggest. Please clarify.

---

## Author Comment (AC1) · 6 Mar 2018

Dear Dr Henderson,

We thank the two anonymous reviewers for their very constructive reviews of our paper "The 226Ra-Ba relationship in the North Atlantic during GEOTRACES-GA01". We have considered all their comments and made the modifications in the manuscript. We think that the scientific results are now better exposed in the revised manuscript. In the following, we answer to each comment of the two reviewers. The referee comments are in blue and our answers are in black.

Please also note the supplement to this comment:

[Figure]

https://www.biogeosciences-discuss.net/bg-2017-478/bg-2017-478-AC1-supplement.pdf

---

## Author Response (AR2)

Dear Dr Henderson,

We thank the two anonymous reviewers for their very constructive reviews of our paper "The $^{226}$Ra-Ba relationship in the North Atlantic during GEOTRACES-GA01". We have considered all their comments and made the modifications in the manuscript. We think that the scientific results are now better exposed in the revised manuscript. In the following, we answer to each comment of the two reviewers. The referee comments are in blue and our answers are in black.

Please note that a new OMP analysis has been performed due to the revision of the paper by Garcia Ibanez et al. in the GEOVIDE special issue. Temperature and salinity of LSW and ISOW have been adjusted to match the salinization observed in recent years. Some nutrient data have been corrected. This slightly impacts $^{226}$Ra and Ba anomalies mostly in the west European Basin (500 m - 1500 m) and near Greenland. Corrections have been made in the text and figures accordingly. These changes are minor and do not impact the discussions nor the conclusions of the manuscript.

Anonymous Referee #1

1. Do seawater 226Ra and Ba behave similarly in the ocean?
The strong correlation between seawater 226Ra and Ba has been commonly used as a result of their similar chemical behaviour. Such an argument is perhaps not as convincing as it used to be since we have seen many pieces of evidence that 226Ra and Ba have different sources and sinks, and can behave differently in different regions (as discussed in Section 4.1 in the manuscript). This is a similar argument for Ba-Si; these two elements seem to be strongly correlated, despite the fact that they have such different chemical cycles in the ocean. Any two elements can be correlated to some extent. However, without fully understanding their removal or regeneration mechanisms, such correlation could also be a coincidence. I think the authors should make that clear in the manuscript, particularly in abstract, conclusion and Section 4.1.

$^{226}$Ra and Ba are chemical analogues since these two elements are alkaline earth element. Even if $^{226}$Ra and Ba enter in the ocean via different sources, $^{226}$Ra and Ba have similar internal cycles in the ocean due to substitution mechanisms (most likely in the same particulate phases) and are then released at depth by dissolution of the settling particles, both elements being redistributed by the global circulation. The OMP shows that away from these sources and sinks $^{226}$Ra and Ba are mostly coupled at intermediate depth and in the interior of the basins, which is in agreement with the view that $^{226}$Ra and Ba are mostly coupled in the ocean. However, our observations led us to the conclusion that "our study also provides evidence of significant decoupling between $^{226}$Ra and Ba", a pattern that is observed close to areas where either $^{226}$Ra or Ba enter into the ocean.

Concerning the Si-$^{226}$Ra (Ba) correlation, it can indeed not be excluded that the relationship is a coincidence, since Si and $^{226}$Ra (Ba) are not chemical analogues. This relationship indeed needs to be better understood. The data collected in the framework of the GEOTRACES program may help to tackle this point.

In the aim to clarify the points mentioned above, we made the following changes in the manuscript:

- Abstract:
Previous version:

$^{226}$Ra and Ba are strongly correlated along the GA01 section, a pattern that reflects their similar chemical behavior."

New version: "$^{226}$Ra and Ba are strongly correlated along the GA01 section, a pattern that **may** reflect their similar chemical behavior." P1 L23-24

Previous version: "However, regions where $^{226}$Ra and Ba displayed non-conservative behaviors were also identified, mostly at the ocean boundaries (seafloor, continental margins, and surface waters)."

New version: "However, regions where $^{226}$Ra and Ba displayed non-conservative behaviors **and in some case decoupled** behaviors were also identified, mostly at the ocean boundaries (seafloor, continental margins, and surface waters)." P1 L34-35

Previous version: "In the upper 1500 m, deficiencies in $^{226}$Ra and Ba are likely explained by their incorporation in planktonic siliceous shells, or in barite ($BaSO_4$) (Bishop, 1988)."

New version: "In the upper 1500 m of the West European Basin, deficiencies in 226Ra and Ba are likely explained by their incorporation in planktonic calcareous and siliceous shells, or in barite (BaSO4) **by substitution or adsorption mechanisms**." P2 L1-3

- Introduction:

Previous version: "Similarly, strong correlations were also found between Ba-Si (silicate) and $^{226}$Ra-Si."

New version: "Similarly, strong correlations were also found between Ba-Si (silicate) and $^{226}$Ra-Si **although no obvious process links $^{226}$Ra or Ba with Si.**" P3 L4-6

- Section 4.1:

Previous version: "This pattern indicates that $^{226}$Ra and Ba behave similarly in the ocean."

New version: "This pattern indicates that $^{226}$Ra and Ba **may** behave similarly in the ocean." P10 L12-13

Previous version: "Since $^{226}$Ra and Ba are incorporated in settling particles such as calcareous and siliceous shells, or barite ($BaSO_4$), and are then released at depth following the dissolution of these particles"

New version: "Since $^{226}$Ra and Ba are incorporated in settling particles such as calcareous and siliceous shells or barite ($BaSO_4$) **by substitution or adsorption mechanisms** (Bishop, 1988; Dehairs et al., 1980; Lea and Boyle, 1989, 1990) and are then released at depth following the dissolution of these particles" P10 L13-14

Previous version: "The link between $^{226}$Ra, Ba and Si has been shown to reflect parallel dissolved-particulate interactions between barite and biogenic silica (Bishop, 1988; Chung, 1980; Jacquet et al., 2005, 2007; Jeandel et al., 1996), the main carrier of Ra in the ocean remains an open question."

New version: "The link between 226Ra, Ba and Si has been shown to reflect parallel dissolved-particulate interactions between barite and biogenic silica (Bishop, 1988; Chung, 1980; Jacquet et al., 2005, 2007; Jeandel et al., 1996); the main carrier of Ra in the ocean, however, remains an open question. **The oceanic Ba-Si and 226Ra-Si relationships may thus be the result of the interaction between ocean biogeochemistry and the water mass transport.**" P10 L24-28

- Conclusion:

Previous version: Finally, $^{226}$Ra and Ba are removed from the upper water column, primarily due to biological mediated processes such as incorporation of $^{226}$Ra and Ba into barite (BaSO$_4$) that are presumably formed following the decay of settling organic matter and/or adsorption onto diatom frustules, a mechanism that would explain the $^{226}$Ra-Ba-**Si** relationship reported here.

New version: "Finally, $^{226}$Ra and Ba are removed from the upper water column, likely primarily due to biological mediated processes such as incorporation of $^{226}$Ra and Ba into barite (BaSO$_4$) that are presumably formed following the decay of settling organic matter and/or adsorption onto diatom frustules, a mechanism that would explain the $^{226}$Ra-Ba relationship reported here. **Similarly, strong correlations were also found between Ba-Si and $^{226}$Ra-Si although no obvious process links $^{226}$Ra or Ba with Si, except maybe for the adsorption of Ba and ($^{226}$Ra) onto diatom frustules. It cannot be excluded, however, that the observed Ba-Si and $^{226}$Ra-Si relationships may result from the spatial coherence of different carriers overprinted by hydrodynamics."** P21 L5-12

2. OMP analysis

One thing that concerns me a little is the uncertainty and accuracy of the OMP analysis. In Section 4.2, the authors mentioned that the OMP analysis suggests that there are 41-66% of MW present between 1000-1600m at Station1 and 21, 60% of ISOW between 2700-3000 m in Station32, and 54% of ISOW between 2100 and 3000m in Station69 and 77. Both MW and ISOW are identified with a relatively low 226Ra/Ba ratio (1.7-1.8, Table S1), compared to 2.0-2.3 in other end-members. So in theory, I would expect to see low 226Ra/Ba in these locations. However, apart from Station69, most of the stations mentioned above either show no change in 226Ra/Ba or show an even higher ratio (i.e. Station32). It would be helpful to show the OMP estimate of 226Ra/Ba in Fig.8 as well. I am also worried that if the estimate of OMP is correct, the absence of low 226Ra/Ba in the intermediate waters (e.g. Station21, 1000-1600m), where >50% was expected to be MW, would suggest that 226Ra/Ba is perhaps not as conservative as we thought even in the intermediate waters.

At the locations mentioned in the comment, where MW (Stations 1 and 13) or ISOW (Stations 69 and 77) are the dominant water masses, the measured $^{226}$Ra/Ba ratios are $2.1 \pm 0.1$ and $2.2 \pm 0.1$dpm µmol$^{-1}$, respectively. At these two locations, the conservative $^{226}$Ra/Ba ratios would be 1.9 dpm µmol$^{-1}$, as deduced from the OMP analysis. The error of the estimation of the conservative $^{226}$Ra/Ba ratios is estimated at 0.3 dpm µmol$^{-1}$. So, we may indeed expect lower $^{226}$Ra/Ba ratios at these stations, but the measured $^{226}$Ra/Ba ratios are actually in agreement with the conservative ratio, considering the errors bars. (See the table below)

| Location | Main Water Masse | $^{226}$Ra/Ba $_{Measured}$ (dpm µmol$^{-1}$) | $^{226}$Ra/Ba$_{Conservative}$ (dpm µmol$^{-1}$) |
|---|---|---|---|
| Stations 1 and 13 | MW | $2.1 \pm 0.1$ | $2.0 \pm 0.3$ |
| Station 32, 69 and 77 | ISOW | $2.2 \pm 0.1$ | $1.9 \pm 0.3$ |

Due to the relatively high errors associated with the estimate of the conservative $^{226}$Ra/Ba ratios determined using OMP analysis, it is difficult to compare the conservative $^{226}$Ra/Ba ratios with the measured $^{226}$Ra/Ba ratios to provide clear trends. That is why the conservative $^{226}$Ra/Ba ratios deduced from the OMP analysis have not been plotted on fig. 8. However, now the error bars appears on $^{226}$Ra/Ba vertical profiles (in fig 8), which was not the case in the first version of the manuscript.

**3. Removal of 226Ra and Ba**

The authors suggest that phytoplankton blooms in the Labrador Sea may explain the negative anomalies of 226Ra and Ba in the areas. I think this statement requires more clarification and work. At least from the Ba point of view, we know that Ba removal (barite formation) is not a direct function of POC flux. In this case, the Labrador Sea (Station60, 64, and 69) shows nearly 10% Ba deficiency in water columns from 200 to 1500 m, according to the OMP analysis (Fig.8d). Such a huge deficiency for a relatively long-residence-time element is unlikely to be explained by seasonal phytoplankton blooms.

Now, due to the new OMP analysis, $^{226}$Ra and Ba the anomalies are lower than in the previous version near Greenland and in the Labrador Sea. The "huge deficiency" as qualified by the reviewer is not present anymore. Therefore the phytoplankton explanation appears now more realistic.

**4. The uncertainty of estimated sediment 226Ra flux**

In Section 4.4, equations (1) and (2) suggest that the uncertainty of sediment 226Ra flux comes from A (positive anomaly), T (transport rate) and S (surface area). At the moment, a 20% error in the calculated 226Ra flux does not include the 33% error from the transport rate. I think the authors should consider the error propagation from each component in the flux calculation.

This is right. Error propagation for transport has now been considered.

Previous version: "The calculated FsedRa is $14.8 \pm 3.1 \ 10^{-3}$ dpm cm$^{-2}$ y$^{-1}$"

New version: "The calculated $Fsed_{Ra}$ is $14.8 \pm \textbf{6.9} \ 10^{-3}$ dpm cm$^{-2}$ y$^{-1}$" P18 L17

**5. How much tolerance does seawater 226Ra/Ba need to be used as conservative tracers (clock) to chronometer the thermohaline circulation?**

Despite that the authors suggest that 226Ra and Ba in intermediate waters are mostly conservative, the readers would not know how sensitive the seawater 226Ra/Ba is as a tracer to chronometer the thermohaline circulation. For example, based on our current understanding of the time for the global ocean circulation, how much decay 226Ra/Ba is expected? Considering that there might still be some non-conservative behaviour (~30%) involved during the ocean basin scale mixing, how much tolerance does seawater 226Ra/Ba have if we want to use it for such an application? The authors could perhaps include some of this discussion in the paper.

The thermohaline circulation time scale has been estimated to be ca. 1000 years from the North Atlantic to the North Pacific using the $^{14}$C/$^{12}$C ratio as a chronometer (Broecker, 1979).

If we use the $^{226}$Ra/Ba ratio to estimate the thermohaline circulation time scale, we can indeed consider the mean $^{226}$Ra/Ba ratio of 2.2 dpm μmol$^{-1}$ for the North Atlantic (between 200 and 800 m at station 69, where the deep waters are formed) and a $^{226}$Ra/Ba ratio of 1.7 dpm μmol$^{-1}$ in the Northeast Pacific (Chung, 1980). Using these two $^{226}$Ra/Ba ratios, the thermohaline circulation time scale is estimated at $616 \pm 197$ years. However, this estimation does not consider the non-conservative behavior for $^{226}$Ra and Ba : along the path to the North Pacific, the deep water masses enter in contact with margins or deep sediments, where they incorporate $^{226}$Ra (and potentially Ba), which is expected to impact the $^{226}$Ra/Ba ratio. The inputs of $^{226}$Ra and Ba near the seafloor along the thermohaline circulation path should be better constrained in order to use the $^{226}$Ra/Ba ratio as a chronometer of the thermohaline circulation.

Previous version: This indicates that the distributions of [226]Ra and Ba at these intermediate depths are largely governed by water mass transport and mixing. The use of the [226]Ra/Ba ratio as a clock to chronometer the thermohaline circulation may thus be relevant when studying water masses at these intermediate depths.

New version: "The absence of a stable isotope for radium led geochemists to consider Ba as a stable analog for [226]Ra because [226]Ra and Ba display a similar chemical behavior, with the aim of using the [226]Ra/Ba ratio as a chronometer for the thermohaline circulation. This study confirms that [226]Ra and Ba behave similarly in the ocean interior away from external sources, both elements being predominantly conservative in the studied area over distances of the order of a few thousands of km. However, this study also highlights regions where [226]Ra and Ba deviate from a conservative behavior, an important consideration when considering the balance between the large-scale oceanic circulation and biological activity over long time scales. Decoupling between [226]Ra and Ba has been observed, in most cases at the ocean boundaries as the result of dissolved [226]Ra and Ba external sources. In addition, suspended particle dissolution may differently impact the dissolved [226]Ra and Ba content of intermediate and deep waters (as shown for the NEADWl); such process would therefore potentially modify their [226]Ra/Ba ratios and would complicate the use of this ratio as a chronometer. Inclusion of the different sources and sinks and particle/dissolved interactions in global ocean models should help to refine the use of the [226]Ra/Ba ratio as a clock to chronometer the thermohaline circulation, as was proposed several decades ago during the GEOSECS program." P21 L20-34

Minor comments:
P3L33 R/V Pourquoi Pas?, the question mark seems to be a typo.
The name of the R/V is indeed "Pourquoi pas ?" with the question mark. This is not a mistake.

P5L7-8 I understand that, in most of the cases, the particulate Ba is low (<1% of the total Ba). I think this needs to be checked for samples in nepheloid layers and hydrothermal plumes as well.
Along the GEOVIDE section, below 1000 m, particulate Ba represents less than 1 % of the total Ba. One exception (bottom of station 32) has been noticed where particulate Ba represents 1.3 % of the total Ba due to the presence of a nepheloid layer.
The particulate Ba data along GEOVIDE are discussed in "Particulate barium tracing significant mesopelagic carbon remineralisation in the North Atlantic" by Lemaitre et al. in this special issue.
Previous version: The Ba measurements presented here are the sum of dissolved Ba and a very small fraction (generally <1 % of total Ba) of particulate Ba released from the samples as a result of the acidification step.
New version: "The Ba measurements presented here are the sum of dissolved Ba and a very small fraction (generally <1 % of total Ba, **along GEOVIDE up to 1.3 % at the bottom of station 32 in a nepheloid layer; Lemaitre et al., this issue**) of particulate Ba released into the dissolved phase as a result of the acidification step." P5 20-22

P13L25-28, some references are required for this statement.
We agree that a reference should be added:
New version: "This pattern is different to that observed in the West European Basin, a discrepancy that may be explained by the different sediment composition in the two regions

and/or by the different residence time of deep waters in contact with deep-sea sediments **(Chung, 1976)**" P15 L5-7

P16L2, would not Ba be scavenged by Mn oxides as well?
We agree that Ba would be scavenged by Mn oxides, likely in a similar manner as $^{226}$Ra.

Previous version: "Radium-226 – although it is much less particle-reactive than $^{230}$Th – may also be scavenged by resuspended particles near the seafloor and may adsorb onto the surfaces of Mn oxides (Moore and Reid, 1973)."
New version: "Radium-226 – although it is much less particle-reactive than $^{230}$Th – **and Ba** may also be scavenged by resuspended particles near the seafloor and may adsorb onto the surfaces of Mn oxides (Moore and Reid, 1973)." P17 L20-22

Fig.6. It would be much more clear if AABW and NEADWI are labeled on the figures to show the transformation of water properties.
This is right. The labels "AABW" and "NEADWl" now appear on the figure 6.

Fig.9. This figure shows some interesting observations. One thing worthy of notice is that the trend is not a linear relationship. I think that there are many reasons (sediment water contact residence time, water age, and scavenging, etc.) to explain why sediment 226Ra flux and bottom water 226Ra concentration are not linearly correlated. This may be something interesting that the authors can point out to their readers in the discussion.
The number of data is actually relatively small here and it is difficult to conclude if it is indeed a linear trend or another trend. All the data from this figure, except for the GEOVIDE value, were reported by Cochran (1980) who discussed the reasons why $^{226}$Ra activities increase in bottom waters in relationship with increasing Ra flux from the seafloor. Because it has already been discussed by Cochran (1980) we did not want to discuss this relation in this paper. But it is true that additional data would be useful to better constrain this relationship.

Cochran (1980) indeed concluded that the observed increase could be due to:
- The residence time of the deep water in each ocean basin. Thus, the deep Atlantic, with a relatively short residence time and a low 226Ra flux from the bottom has low 226Ra concentrations. In the northeast Pacific, the geographic variation in near-bottom 226Ra is closely matched by the variation in 226Ra flux from the bottom.
- The distribution of biogenic opal in north Pacific sediments is geographically concentrated near the equator.

Anonymous Referee #2

P3 Line 5: explain parallel carriers

The term "parallel carrier" was not clear, "different" carriers are now reported.

Previous version: "Rather, the observed relationships may result from the spatial coherence of parallel carriers overprinted by hydrodynamics."
New version: "Rather, the observed relationships may result from the spatial coherence of **different carriers (barite, opal and carbonate)** overprinted by hydrodynamics." P3 L11-13

The description of the water masses and circulation as shown in Figure 1 could be more accurate: We made changes according to the comments (see below)

P 6, Line 24: from Fig. 1 it appears that stations 1 to 26 cover the section between the Iberian Peninsula and the Rockall Through, and not till Reykjanes Ridge as indicated in the text
Previous version: "Three main water masses were found in upper waters (<1000 m) in the investigated area (Fig.3). First, the Central Waters occupied the upper eastern part of the GA01 section from the Iberian Peninsula to the Reykjanes Ridge."
New version: "Three main water masses were found in upper waters (<1000 m) in the investigated area (Fig.3). First, the Central Waters occupied the upper eastern part of the GA01 section from the Iberian Peninsula to **the Rockall Through**." P7 L5-6

P 6, Lines 27-28: From Figure 1 splitting of the NAC occurs west of the Mid Atlantic ridge, not at the MAR as indicated in text.

Previous version: "The NAC flows eastward from the Grand Banks of Newfoundland, splitting into four branches of the Mid-Atlantic Ridge (MAR), while incorporating local water masses (Fig.1)."
New version: "The NAC flows eastward from the Grand Banks of Newfoundland, splitting into four branches **west** of the Mid-Atlantic Ridge (MAR), while incorporating local water masses (Fig.1)." P7 L9-12

P6, Lines 30-31: the southern branch of the NAC appears to flow into the Iberian abyssal plain, rather than into the West European basin.
We agree with this remark. The arrow on the figure has been changed.

P9, Lines 17-18: positive intercept on Ba axis is explained as either resulting from larger rivers input relative to 226 Ra and larger 226Ra input relative to Ba at the sediment interface; it is not clear how the latter situation may lead to a positive intercept on the Ba axis.
We agree with this remark. The largest 226Ra input from the sediment relative to Ba cannot explain the positive interface. We changed the sentence according to this remark.

Previous version: "This positive intercept may be the result a greater input of 226Ra relative to Ba close to bottom sediments and a larger riverine Ba input relative to 226Ra."
New version: "This positive intercept may be the result a larger riverine Ba input relative to $^{226}$Ra (Ku and Luo, 1994)." P10 L8-9

P9: Lines 20-21: slope of regression curves 226Ra-Ba. Alert the reader that units in Fig 5 are not the same as those for the global ocean Ra/Ba ratio.
This information has been added in the figure caption to alert the reader.

Previous version: "Figure 5: Relationships between $^{226}$Ra and Ba (red dots) and between $^{226}$Ra and Si(OH)$_4$ (blue dots) along the GA01 section in the North Atlantic. The best linear fit for the two plots is also reported (R=0.93 for the two plots)."
New version: Figure 5: "Relationships between $^{226}$Ra and Ba (red dots) and between 226Ra and Si(OH)4 (blue dots) along the GA01 section in the North Atlantic. The best linear fit for the two plots is also reported (R=0.93 for the two plots). **The slopes of the relationships between $^{226}$Ra-Ba and between $^{226}$Ra-Si(OH)4 are expressed in 10-2 dpm nmol-1 and in 10-2 dpm μmol-1, respectively**." P22 L17-18

P9 Line 25: change sentence ". . . small fractionation between Ra and Ba during these processes" to "indicates the fractionation between Ra and Ba during these processes is small".
Right, we made the change as follows :
Previous version: "the constant 226Ra/Ba ratio suggests small fractionation between 226Ra and Ba during these processes."
New version: "the constant 226Ra/Ba ratio suggests that the fractionation between 226Ra and Ba during these processes is small". P10 L16-17

P10 Lines 5 to 15: Some words of explanation are needed when discussing varying Ra to Si ratios between the major basins. A look at the Sarmiento et al. Nature paper of 2007 could be helpful.
Previous version: "In contrast to the $^{226}$Ra-Ba relationship, the slope of the $^{226}$Ra-Si(OH)$_4$ relationship during GEOSECS exhibited significant spatial variability from one oceanic basin to the other (Li et al., 1973)"
New version: "In contrast to the $^{226}$Ra-Ba relationship, the slope of the $^{226}$Ra-Si(OH)$_4$ relationship during GEOSECS exhibited significant spatial variability from one oceanic basin to the other (Li et al., 1973). **First, $^{226}$Ra and Si are not chemical analogues, as is the case for $^{226}$Ra and Ba. Second, the variability observed in the $^{226}$Ra-Si(OH)$_4$ relationship may also be related to the large variability of the Si(OH)4 distribution which is mostly governed by the preformed nutrient concentrations of waters feeding into the main thermocline from surface waters of the Southern Ocean (Sarmiento et al., 2007)." P10 L29 - P11 L1**

P10 Line23: "These waters then sink and circulate northward into the Atlantic Ocean" specify that these 'waters' are mainly AAIW and also to some extent SAMW.
This part have been changed to:
"The NEADWl is mainly formed from waters with a southern origin (Read, 2000). South of the Antarctic Convergence, the surface waters contain high $^{226}$Ra activities from the upwelling of deep waters enriched in $^{226}$Ra associated with the circumpolar current (Ku and Lin, 1976). The convection of these surface waters leads to the formation of the 226Ra-rich AABW that circulates northward into the Atlantic Ocean." P11 L16-20

P10, Line 25 (Fig. 6) and onwards: What explains this abrupt decrease of 226Ra between Eq. and 11°N?
What is the rationale to consider these meridional profiles as a continuum, or to what extent is this reconstructed meridional section of 226Ra and the other variables depending on the basin they originate from (i.e. west or east Atlantic Basin)? This needs to be clarified.

Here we follow the path of the AABW, first in the West Atlantic Basin from south to north. Second, the AABW crosses the ridge (11°N) from west to east and changes into the NEADWl due to the mixing with northern water masses while crossing the ridge. Then the NEADWl keeps going north in the East Atlantic Basin.

In figure 9 we used available data as close as possible of the core of the water masses according to the circulation and characteristics of the water masses.

In the aim to clarify, some major modifications have been made (note that Fig.6 is now Fig.8):

Previous version: "Broecker et al., (1976) showed that the decrease in the $^{226}$Ra activities from south to north is produced by the mixing of the AABW and bottom waters of northern origin while crossing the Mid-Atlantic Ridge from the West to the East Atlantic Basin. Figure 6 was computed combining GEOSECS and TTO data ($^{226}$Ra, Si(OH)$_4$, salinity and temperature) gathered in the AABW that travels northward between 60°S and 40°N in the West Atlantic Basin. The same data ($^{226}$Ra, Si(OH)$_4$, salinity and temperature) determined along GA01 are also reported."

New version: "As mentioned above, the NEADWl—which is the main water mass of the deep West European Basin—is mainly formed from waters with a southern origin (mainly AABW) that are characterized by high $^{226}$Ra and Ba concentrations. However, these southern waters experience a very specific history along their northward transport to the GA01 section that suggests that the high $^{226}$Ra activities (and Ba) of the NEADWl cannot be solely explained by the high $^{226}$Ra activities (and Ba) of these waters of southern origin. In order to explain the positive $^{226}$Ra and Ba anomalies in the deep waters of the West European Basin, we thus need to investigate the fate of $^{226}$Ra and Ba in the waters of southern origin that travel northward and reach section GA01. Figure 8 was computed combining GEOSECS and TTO data ($^{226}$Ra, Si(OH)$_4$, salinity and temperature) associated with the AABW (Spencer, 1972) that travels northward between 60 °S and 40 °N in the West Atlantic Basin. The same data ($^{226}$Ra, Si(OH)$_4$, salinity and temperature) associated with the NEADWl in the East Atlantic Basin and along GA01 are also reported.

Between 60 °S and the equator, the high $^{226}$Ra activities of the AABW are associated with relatively low salinity, low temperature, and high Si(OH)$_4$ (Fig. 8). Then, while crossing the Mid-Atlantic Ridge at the equator and at 11 °N, the AABW undergoes several important transformations: $^{226}$Ra activities and Si(OH)$_4$ concentrations decrease while salinity and temperature tend to increase (Fig. 8). The $^{226}$Ra and Ba endmembers for the NEADWl were chosen at this specific location to coincide both geographically and with the characteristics (salinity, temperature and Si(OH)$_4$) of the NEADWl endmembers used for the OMP analysis (Fig. 8; Fig.S2). This endmember has been chosen far from the GA01 section in the OMP analysis (García-Ibáñez et al., this issue), because between 11 °N and the GA01 section (Fig. 8), salinity, temperature, and Si(OH)$_4$ concentrations display relatively constant trends indicating no major modifications. In contrast, the $^{226}$Ra activities display a significant spatial variability north of 11 °N, and clearly increase towards the north (Fig. 8). This $^{226}$Ra increase is thus decoupled from salinity, temperature, and Si(OH)$_4$), and likely explain the positive anomalies deduced from the OMP analysis in the deep West European Basin (Fig. 7). The specific history of these waters of southern origin (waters initially with a high $^{226}$Ra activity; decrease in the $^{226}$Ra activity at the equator and at 11 °N; new increase of $^{226}$Ra activity north of 11 °N) suggest that the $^{226}$Ra anomalies observed in the West European Basin are explained by inputs of $^{226}$Ra along the northward transport of these waters." P13 L12-P14 L7

Line 32: clarify the meaning of 'coincide'; you mean geographical coincidence or compositional similarity?

We agree that it was not clear. Therefore, we made the following change:

Previous version: "The $^{226}$Ra and Ba endmembers for the NEADWl were chosen at this specific location to coincide with the NEADWl endmembers used for the OMP analysis (Fig.6; Fig.S2)."

New version: "The $^{226}$Ra and Ba endmembers for the NEADWl were chosen at this specific location to coincide **both geographically and with the characteristics (salinity, temperature and Si(OH)$_4$)** of the NEADWl endmembers used for the OMP analysis (Fig. 8; Fig.S2)"P13 28-30

P10 Lines 34-35: sentence unclear. 226Ra activities are high when waters reach 40°N. So they are high compared to the western part of the GA01 section (not the eastern part) .

There was indeed a mistake, the paragraph has been changed in this new version.

Previous version: "these waters are then characterized by high 226Ra activities compared to the waters located in the **eastern** part the GA01 section."

New version: "A striking feature of the GA01 section is that the $^{226}$Ra activities and Ba concentrations are particularly high in the West European Basin below 2000 m (Fig.4), in the the NEADWl. P11 L13-14

P11 Line 13: presence of the MW at about 41% - 66%; % of what? Specify.

In order to clarify this point, we changed the sentence:

Previous version: "At these depths, the OMP analysis confirms the presence of the MW at about 41 % - 66 % (stations 1 and 21; Garcia Ibanez et al., 2017; this issue)."

New version: "At these stations, between 30 and 79 % of the water found at 1000–1600 m is of Mediterranean origin (MW), according to the OMP analysis (Garcia Ibanez et al., 2017; this issue)." P11 L1-2

Note that the percentages have been changed due to the new OMP analysis.

P12 Lines 25- 35: What may cause the absence of a positive anomaly for Si, while the anomaly is strong for Ra ? Is it because the region is dominated rather by coccolithophorids (compared to the western part of the section, where the diatoms dominate), and so sediments may be relatively poor in Si ? Comment please.

Indeed, we sometimes observe decoupling between 226Ra and Si(OH)4. High values of 226Ra are not always accompanied by high Si(OH)4 concentrations (Figure 8). The reasons for such decoupling are unclear. Si(OH)4 may be more conservative than 226Ra in this region (deep West European basin), since we found that significant sedimentary input of 226Ra takes place in this region. The low sedimentary input of Si(OH)4 may be related to the sediments that may be indeed relatively poor in Si in this region.

P14: title of section 4.3.2 lacks a verb ?

Previous version: "$^{226}$Ra removal its relationship with Ba"

New version: "$^{226}$Ra removal **and** its relationship with Ba" P15 L28

P14 Line14: Acantharians are invoked to explain lower Ra/Ba ratios. Is there any direct evidence during the GEOVIDE cruise for the presence of these organisms?

Unfortunately, the presence of Acantharians was not studied during GEOVIDE. However, previous studies reported presence of acantharians in this area, as mentioned below.

We also rely on previous studies that related the low dissolved 226Ra/Ba ratios to the presence of acantharians (van Beek et al., 2007; van beek et al., 2009). Such low dissolved 226Ra/Ba ratios may thus be characteristic (specific isotopic signature) of the presence of this class of organisms.

Previous version: "Previous studies reported the presence of these organisms in the North Atlantic, especially in the Iceland Basin and in the East Greenland Sea (Antia et al., 1993; Barnard et al., 2004; Martin et al., 2010)"

New version: "**The presence of Acantharians was not studied during GEOVIDE.** However, previous studies reported presence of acantharians in this area: previous studies reported the presence of these organisms in the North Atlantic, especially in the Iceland Basin and in the East Greenland Sea (Antia et al., 1993; Barnard et al., 2004; Martin et al., 2010)."P16 L11-14

P14-15: Differentiation of west and east regarding dominance of diatoms (west) vs coccolithophorids (east) as such is not sufficient to explain the 226Ra distribution. Is it known that these two phytoplankton groups accumulate Ra and Ba in a different way? Please comment further.
To our knowledge, it is not clear if these two phytoplankton groups accumulate Ra and Ba in a different way. Ba (and therefore Ra) is incorporated into calcium carbonate (coccolithophorids), since Ba (and Ra) would substitute for Ca. Regarding diatoms, it is not clear how Ba (and Ra) would react with the diatom frustules, but it was proposed that Ba (and Ra) would adsorb onto the frustules, which could promote barite precipitation (Bishop, 1988).
Previous version: "Since Diatoms are the dominant species in the Irminger and Labrador Seas and on the Greenland and Newfoundland margins during GA01 (up to 55 % of the total Chl-a concentration; Tonnard et al., in prep), the diatom frustules may also contribute to the removal of 226Ra and Ba, from the water column in these areas that were characterized by noticeable negative anomalies."
New version: "In the West European Basin, Chl-a was lower in May and June 2014 and coccolithophorids were the dominant species in that area (Tonnard et al., in prep). In these two regions, diatom frustules and coccolithophorids may thus contribute to the removal of $^{226}$Ra and Ba (Bishop, 1988; Dehairs et al., 1980) from the water column in these areas that were characterized by noticeable negative anomalies." P16 L19-L22

P16 Line28: . . . "which is within the range of fluxes" . In fact the calculated flux is clearly larger than the reported fluxes in the literature, so stating that the statement is not appropriate.
The calculated $Fsed_{Ra}$ is $14.8 \pm 6.9 \ 10^{-3}$ dpm cm$^{-2}$ y$^{-1}$. This is within the range of fluxes reported in the literature (considering all oceanic basins):
$1.5 \ 10^{-3}$ dpm cm$^{-2}$ y$^{-1}$ < $Fsed_{Ra}$ < $2.1 \ 10^{-1}$ dpm cm$^{-2}$ y$^{-1}$. Therefore we did not modified this part.

P17 lines 15 – 20: It is not because Ra and Ba contents in particles are much lower than the contents in solution, that particle dissolution should be considered as minor. This depends on turn-over rate of particulate vs dissolved phase.

Yes, we agree with the reviewer and acknowledge that our reasoning was erroneous. We especially thank the reviewer for this comment (in addition to all others). This point has been taken into account in the revised version through a significant additional discussion:

Additional discussion in section 4.4.:

"Alternatively, the dissolution of settling particles could also contribute to the $^{226}$Ra and Ba anomalies observed in the deep waters of the West European Basin. Assuming steady state, we may undertake a mass balance calculation for particulate $^{226}$Ra and Ba in the same box as described above (i.e., box defined between 11 °N and the GA01 section, between stations 1 and 21 –1475 km–and between 3500 m depth and the seafloor; Fig.S5). Particles enter the box from above as settling particles, but also horizontally, carried within the water masses at 11 °N that travel northward. Particles leave the box by different processes (accumulation in the sediment or northward transport by the water masses) or alternatively may dissolve while settling in the box. In the absence of precise information about the particulate $^{226}$Ra and Ba fluxes entering and exiting the box horizontally (i.e. the particulate $^{226}$Ra and Ba concentrations at 11 °N and at the GA01 section), we assume that they are of equal importance and therefore that they cancel each other in the mass balance calculation.

The vertical particulate flux entering the box, from above, can be calculated as follows:

$$F_{Part-x} = Cp_{3500} \times Vs \times S \ (3)$$

where $x$ is either $^{226}$Ra or Ba, $Cp3500$ is either the particulate $^{226}$Ra activities or the particulate Ba concentrations at 3500 m, $Vs$ is the settling speed for suspended particles and $S$ is the horizontal surface area described above (6.21 $10^6$ km$^2$).

We use the value of 0.007 dpm 100 L$^{-1}$ for the mean $^{226}$Ra particulate activity at 3500 m, a value that was reported for the Atlantic Ocean, Sargasso Sea (van Beek et al., 2007) and the value of 0.087 nmol L$^{-1}$ for the mean Ba particulate concentration at 3500 m, a value that was determined along the GA01 section (Lemaitre et al., this issue). We use the settling speeds (Vs) reported for suspended particles in the literature and that typically range from 100 to 1000 m y$^{-1}$ (Bacon and Anderson, 1982; Krishnaswami et al., 1976; Roy-Barman et al., 2002). The $F_{Part}$ thus obtained range from 1.4 $10^6$ to 13.8 $10^6$ dpm s$^{-1}$ for $^{226}$Ra, while the $F_{Part}$ range from 1.7 and 17.2 mol s$^{-1}$ for Ba. Of this total F$_{Part}$, some fraction may dissolve while settling, while the remainder will accumulate in the sediment. This dissolution flux is named F$_{dissolution-x}$, where x is either $^{226}$Ra or Ba. We use the sediment Ba accumulation rates reported by Gingele and Dahmke (1994) in the Atlantic Ocean to calculate the particulate Ba flux that exits the box (F$_{Accumulation-Ba:}$ 2.0 to 13.4 mol s$^{-1}$); hence, by difference the F$_{dissolution-Ba}$ is 0-15.2 mol s$^{-1}$ (Fig.10). This value is of the same order of magnitude of the $F_{Input-Ba}$ needed to explain the Ba anomalies (6.28 mol s$^{-1}$). Therefore, in the case of Ba, the dissolution of settling particles may entirely explain the OMPA-derived anomalies. The sediment $^{226}$Ra accumulation rates can be calculated from the Ba accumulation rates estimated above, using the $^{226}$Ra/Ba ratio determined in sinking particles collected in the Sargasso Sea near the seafloor (i.e., 1.5 dpm μmol$^{-1}$; van Beek et al., 2007). The sediment $^{226}$Ra accumulation flux thus calculated, F$_{Accumulation-Ra}$, is 2.9 $10^6$-19.6 $10^6$ dpm s$^{-1}$, leading to F$_{dissolution-Ra}$ of 0-10.9 $10^6$ dpm s$^{-1}$ (Fig.10). Therefore, F$_{dissolution-Ra}$ cannot account for more than 37 % of the required $^{226}$Ra flux ($F_{Input-Ra}$). This implies that even if the settling speed is high (1000 m y$^{-1}$; high turnover of the particles), the particle dissolution cannot account for the entire $F_{Input-Ra}$. The remaining (minimum of 63 %) therefore has to be sustained by $^{226}$Ra diffusion from the sediments.

While the above calculations have to be taken with caution given the numerous assumptions in the mass balance model, overall they suggest that the $^{226}$Ra positive anomalies

observed in the West European Basin may be explained entirely by $^{226}$Ra that diffuses out of the sediments. However, it cannot be excluded that the dissolution of settling particles also contributes to the $^{226}$Ra enrichment. In contrast, the Ba positive anomalies may be explained either by the diffusion of Ba from sediment or by the dissolution of settling particles or by a combination of these two processes. These conclusions are in line with the current knowledge about $^{226}$Ra and Ba sources in the deep ocean (Broecker et al., 1970; Chan et al., 1976, 1977; Ku et al., 1980)."

P17 Lines 31-32: Potential for 226Ra/Ba ratio to be used as a clock for THC. From the data here in the North Atlantic there is but poor evidence that with time 226Ra activity relative to Ba is decreasing. Also globally the Ra/Ba ratio appears to stick closely to the 2.2 value. Can you comment on what the effective perspective is for the use of this ratio as a chronometer ? Is it a question of insufficient precision on the present day data ? If so what needs to be done ? A few lines discussing the status of this issue would be welcome.

The thermohaline circulation time scale has been estimated to be ca. 1000 years from the North Atlantic to the North Pacific using the $^{14}$C/$^{12}$C ratio as a chronometer (Broecker, 1979).

If we use the $^{226}$Ra/Ba ratio to estimate the thermohaline circulation time scale, we can indeed consider the mean $^{226}$Ra/Ba ratio of 2.2 dpm µmol$^{-1}$ for the North Atlantic (between 200 and 800 m at station 69, where the deep waters are formed) and a $^{226}$Ra/Ba ratio of 1.7 dpm µmol$^{-1}$ in the Northeast Pacific (Chung, 1980). Using these two $^{226}$Ra/Ba ratios, the thermohaline circulation time scale is estimated at $616 \pm 197$ years. However, this estimation does not consider the non-conservative behavior for $^{226}$Ra and Ba : along the path to the North Pacific, the deep water masses enter in contact with margins or deep sediments, where they incorporate $^{226}$Ra (and potentially Ba), which is expected to impact the $^{226}$Ra/Ba ratio. The inputs of $^{226}$Ra and Ba near the seafloor along the thermohaline circulation path should be better constrained in order to use the $^{226}$Ra/Ba ratio as a chronometer of the thermohaline circulation.

Previous version: This indicates that the distributions of $^{226}$Ra and Ba at these intermediate depths are largely governed by water mass transport and mixing. The use of the $^{226}$Ra/Ba ratio as a clock to chronometer the thermohaline circulation may thus be relevant when studying water masses at these intermediate depths.

New version: "The absence of a stable isotope for radium led geochemists to consider Ba as a stable analog for $^{226}$Ra because $^{226}$Ra and Ba display a similar chemical behavior, with the aim of using the $^{226}$Ra/Ba ratio as a chronometer for the thermohaline circulation. This study confirms that $^{226}$Ra and Ba behave similarly in the ocean interior away from external sources, both elements being predominantly conservative in the studied area over distances of the order of a few thousands of km. However, this study also highlights regions where $^{226}$Ra and Ba deviate from a conservative behavior, an important consideration when considering the balance between the large-scale oceanic circulation and biological activity over long time scales. Decoupling between $^{226}$Ra and Ba has been observed, in most cases at the ocean boundaries as the result of dissolved $^{226}$Ra and Ba external sources. In addition, suspended particle dissolution may differently impact the dissolved $^{226}$Ra and Ba content of intermediate and deep waters (as shown for the NEADWl); such process would therefore potentially modify their $^{226}$Ra/Ba ratios and would complicate the use of this ratio as a chronometer. Inclusion of the different sources and sinks and particle/dissolved interactions in global ocean models should help to refine the use of the $^{226}$Ra/Ba ratio as a clock to chronometer the thermohaline circulation, as was proposed several decades ago during the GEOSECS program." P14 L23-34

P18 Line 1-2: The absence of Ba enrichment in deep waters in the western part of the section, while Ra is enriched is not really discussed. What could be the possible mechanisms?

A possible explanation is mentioned in the section 4.3.1. :

"The $^{226}$Ra positive anomalies observed at the stations mentioned above are thus best explained by the diffusion of $^{226}$Ra from the sediment. However, these latter stations do not exhibit a positive Ba anomaly and Ba tends to be conservative. Consequently, the $^{226}$Ra/Ba ratios in the deep waters of these stations are often significantly higher than the mean GEOSECS value (stations 21, 32, 38, 60, 64; Fig. 7). This pattern is different to that observed in the West European Basin, a discrepancy that may be explained by the different sediment composition in the two regions, by the different residence time of deep waters in contact with deep-sea sediments (Chung, 1976) and/or different role played by suspended particles dissolution." P15 L 1- 9

Figures and tables:

Changes have been made in the tables and figures according to the remarks.

Legends should detail all acronyms (water masses) shown or listed.

Added as Table 1

Table S2:

It would be useful to indicate the CTD number of the casts.

CTD numbers have been added

Why not add also the Si(OH)4 profiles ?

The SI(OH)$_4$ data will be published in another paper of the special issue. The vertical profiles are shown in the supplementary material.

Giving depth values with a decimal seems not realistic given natural variabilities due to waves, swells ..

Decimals have been removed

Why are activity and concentration data not expressed per unit weight (Kg) rather than per unit volume, as is standard practice?

The data are not expressed per unit of weight in the aim to calculate directly the $^{226}$Ra/Ba ratio, Ba being expressed in nmol L$^{-1}$.

Also for Ba, the indicated 1.5% precision implies the numbers should be rounded to the first decimal.

These data have been changed to one decimal.

Some numbers may be suspicious: examples are station 21: 226Ra = 24.87 at 4176m; temperature at 794m; station 32: 226Ra at 794m; please check.

These points have been checked and changes have been made when necessary.

Station 60: Why is the depth range not continuous? Are the data from two different CTD casts ? This is what Sal and Temp suggest. Please clarify

Thanks. It was not the same station, the error has been corrected.